# ON THE FRAGILITY OF GRAPH BACKDOOR DEFENSES: A ROBUST STRATEGY VIA LAYER-WISE FEATURE DIVERGENCE

## ABSTRACT

Recent studies have revealed the high susceptibility of GNNs against backdoor attacks, which poses a significant threat to their practical applications. In order to deal with the threats posed by backdoors, a series of targeted defense measures have been proposed, which have effectively alleviated the harm of backdoor attacks to a certain extent. However, do these methods really completely eliminate the threat of backdoors? Inspired by related research in the DNN field, we conduct the first systematic robustness analysis of backdoor defenses in the GNN domain. Our experiments reveal that even fine-tuning the defense model for only five epochs with a small fraction of poisoned data can cause a sharp resurgence in its ASR, indicating that residual backdoor features persist and can be readily reactivated. Recognizing the unique message-passing paradigm in GNNs, we leverage Layer-wise Linear Feature Connectivity (LLFC) to uncover the root cause of this pronounced fragility in current graph backdoor defenses. To enhance the robustness of these defenses, we also propose a novel strategy termed **Layer-wise Feature Divergence (LFD)**, which forces the defense model to diverge from the original backdoor model by maximizing the distance between their respective layer-wise features during retraining. Extensive experiments demonstrate that LFD significantly enhances the robustness of defense models, achieving state-of-the-art performance in defense capabilities while maintaining high accuracy on clean data.

## 1 INTRODUCTION

Graph neural networks (GNNs) have emerged as powerful tools for learning from graph-structured data and have achieved remarkable success in diverse domains, including social network analysis (Yang et al., 2021; Sankar et al., 2021; Cao et al., 2024), molecular property prediction (Guo et al., 2021; Li et al., 2022a), and recommendation systems (Kumar et al., 2022; Wang et al., 2024; Amara et al., 2025). The strength of GNNs lies in their message-passing mechanism (Kipf, 2016; Feng et al., 2022), which allows nodes to iteratively aggregate neighborhood information and learn expressive representations that capture both node attributes and local topology.

Despite these successes, the security of GNNs has drawn increasing concern. Recent studies demonstrate that GNNs are highly vulnerable to backdoor attacks (Yang et al., 2024a; Alrahis et al., 2023; Ding et al., 2025). In such attacks, adversaries inject carefully crafted triggers, such as small subgraphs with specific structures, into a subset of training data and relabel them to a target class. The resulting backdoor model behaves normally on clean inputs, but misclassifies any input containing the trigger into the attacker's target class. Due to their stealthiness and efficiency, such attacks pose serious risks for deploying GNNs in safety-critical areas, including medical diagnosis and fraud detection (Dou et al., 2020; Liu et al., 2021).

To mitigate these threats, various defense strategies have been proposed (Min et al., 2023; Wan et al., 2025). These include data preprocessing methods that filter poisoned samples (Dai et al., 2023), model modification approaches such as pruning suspicious components (Li et al., 2021), and post-purification defenses that fine-tune models on clean data to forget backdoors (Rosenfeld et al.,

2020; Zhang et al., 2025). Empirically, these defenses often reduce the attack success rate (ASR) close to zero, giving the impression of security.

However, a critical question remains: Do these defenses truly eliminate backdoor features, or do they merely suppress them? Recent work on deep neural networks (DNNs) has shown that many defenses achieve only superficial security—their backdoors can be easily reactivated by fine-tuning with a small amount of poisoned data (Neyshabur et al., 2020; Lubana et al., 2023; Min et al., 2024). This issue, however, has not been systematically studied in the graph domain.

In this paper, we provide the first systematic analysis of the robustness of graph backdoor defenses after purification. Through carefully designed retraining attacks on purified GNNs, we reveal that mainstream defenses exhibit severe robustness flaws: while they initially achieve low ASRs, these quickly rebound to dangerous levels after only brief retraining with a small number of poisoned samples. This finding highlights that residual backdoor knowledge persists, leaving models vulnerable to secondary infection. To explain this phenomenon, we introduce Layer-wise Linear Feature Connectivity (LLFC) proposed by Zhou et al. (2023), a tool for analyzing similarities in the feature space of different models. Our analysis shows that *secure solutions* produced by current defenses remain highly similar to backdoored models in the hierarchical feature space, despite appearing safe in parameter space. Building on this insight, we propose **Layer-wise Feature Divergence (LFD)**, a new training strategy that regularizes purification to maximize divergence from backdoored models in feature space, thereby achieving more robust defenses. In summary, our main contributions are:

- We are the first to systematically reveal the superficial security problem of existing graph backdoor defenses, demonstrating that residual backdoor features can be easily reactivated.

- We introduce LLFC as an analytical tool to explain this vulnerability from the feature-space perspective, offering a new lens for understanding backdoor defenses.

- We propose LFD, a robust defense training strategy that explicitly maximizes feature-space divergence from backdoored models, substantially improving defense robustness.

- Extensive experiments across multiple datasets and attack scenarios validate our findings. Our method consistently achieves stronger robustness while preserving high benign accuracy and maintaining low ASR.

## 2 RELATED WORK

### 2.1 GRAPH BACKDOOR ATTACKS

Backdoor attacks were first extensively studied in computer vision (Li et al., 2022b). The key idea is to implant attacker-controlled triggers into the training data, thereby creating hidden malicious behaviors in the model. Recently, this paradigm has been successfully extended to graph learning (Yang et al., 2024a; Alrahis et al., 2023; Xi et al., 2021), where the richer structure of graph data enables more diverse forms of triggers.

Early works, such as SBA (Zhang et al., 2021) and CBA (Turner et al., 2018), attached randomly generated subgraphs as universal triggers to target nodes. While pioneering, these methods achieved limited attack success rates. To enhance effectiveness, later approaches introduced sample-adaptive triggers. For instance, GTA (Xi et al., 2021) generates customized subgraph triggers for each target node, substantially improving attack success. GCBA (Zhang et al., 2023) extends backdoor attacks to graph contrastive learning, designing attack behaviors aligned with the three stages of contrastive learning.

As defenses advanced, stealthiness became crucial. Early triggers often exhibited feature discrepancies from target nodes (Zhang et al., 2021; Xi et al., 2021; Yang et al., 2024b), violating the common homophily assumption of real-world graphs (i.e., similar nodes tend to connect), making them detectable through anomaly-based defenses. To overcome this, UGBA (Dai et al., 2023) proposed imperceptible triggers by optimizing node features to maximize cosine similarity with targets, improving integration into the host graph. DPGBA (Zhang et al., 2024) further adopted an adversarial learning strategy to generate in-distribution triggers, which closely mimic benign nodes statistically and thus evade detection-based defenses.

## 2.2 GRAPH BACKDOOR DEFENSES

To counter graph backdoor attacks, researchers (Wan et al., 2025) have proposed a range of defense strategies, which can be broadly categorized into data preprocessing and model purification.

**Data preprocessing methods** clean graph inputs before training by detecting and removing suspicious nodes or structures. Prune mentioned in Dai et al. (2023), based on the homophily assumption, calculates node feature similarity and prunes edges connecting dissimilar nodes, thereby disrupting triggers. Other methods treat triggers as anomalies and employ graph outlier detection (OD). For instance, Zhang et al. (2024) trains a graph autoencoder to identify nodes with high reconstruction errors as potential triggers.

**Model purification methods** instead operate post-training to remove backdoor functionality without directly modifying the data. Typical approaches include model pruning, which eliminates neurons or weights associated with backdoor behaviors, and fine-tuning, which retrains backdoored models on a small clean dataset to overwrite backdoor knowledge while preserving benign performance. More recent methods introduce new training principles: RIGBD (Zhang et al., 2025) leverages prediction variance under random edge dropping to design a robust loss, FIP (Karim et al., 2024) exploits loss sharpness and re-optimizes models toward smoother minima using Fisher Information regularization, and DShield (Yu et al., 2025) applies differential learning to defend against both dirty-label and clean-label attacks.

## 2.3 LAYER-WISE LINEAR FEATURE CONNECTIVITY

Linear Mode Connectivity (LMC) is a theoretical tool for studying the relationship between model solutions and the loss landscape (Garipov et al., 2018; Frankle et al., 2020). Two models are said to exhibit LMC if a straight path in parameter space connects them while maintaining low loss along the path, implying convergence to the same flat basin and functional similarity. LMC has also been applied to analyze backdoor defenses. Neyshabur et al. (2020); Min et al. (2024) used it to quantify the distance between purified and backdoored models, showing that more robust purification solutions encounter higher loss barriers along the connecting path, indicating greater deviation from the backdoored solution.

Layer-wise Linear Feature Connectivity (LLFC) generalizes this idea by analyzing connectivity in the feature space across layers (Zhou et al., 2023). Specifically, for two well-connected models, the interpolated model's feature maps at each layer are expected to linearly interpolate between those of the original models. Violations of this property at specific layers highlight functional discrepancies between purified and backdoored models. Compared to the single scalar perspective offered by LMC, LLFC provides fine-grained insights into where and how backdoor features persist within the network (Du et al., 2024).

## 3 PRELIMINARY

In this work, we focus on the inductive task of node-level classification. Let $\mathcal{G} = \{\mathcal{V}, \mathcal{E}, X\}$ denote an original attributed graph, where $\mathcal{V} = \{v_i\}_{i=1}^N$ is the set of $N$ nodes, $\mathcal{E} = \{e_{ij}\}$ is the set of edges, each edge $e_{ij} = \{v_i, v_j\}$ connects nodes $v_i$ and $v_j$, and $X = \{x_1, \cdots, x_N\}$ is the set of attributes of $\mathcal{V}$. The overall topology of the graph $\mathcal{G}$ is represented by the adjacency matrix $A \in \{0, 1\}^{N \times N}$, where $A_{ij} = 1$ if the edge $\{v_i, v_j\}$ is present; otherwise $A_{ij} = 0$. During the training phase, we are provided with a graph $\mathcal{G}_T = \{\mathcal{V}_T, \mathcal{E}_T, X_T\}$. We use $\mathcal{V}_C \subseteq \mathcal{V}_T$ and $\mathcal{V}_P \subseteq \mathcal{V}_T$ to denote the clean node set and the poisoned node set, respectively. Here, a node $v_i \in \mathcal{V}_C$ is clean and its clean label is $y_i$, while a node $v_j \in \mathcal{V}_P$ is poisoned and labeled with the target class $y_t$. The remaining nodes in set $\mathcal{V}_T$ represent the set of unlabeled nodes. During the inference phase, we are provided with a graph $\mathcal{G}_I = \{\mathcal{V}_I, \mathcal{E}_I, X_I\}$. Similarly, a node $v_i \in \mathcal{V}_{I_c}$ is clean, while a node $v_j \in \mathcal{V}_{I_p}$ is poisoned. Follow the inductive setting, $\mathcal{V}_T \cap \mathcal{V}_I = \emptyset$. And the neighbors of node $v_i$ is represented as $\mathcal{N}_i$.

**Threat Model.** During backdoor attack, the attacker aims to inject unnoticeable triggers (usually in the form of a single node or a subgraph) into a small set of target nodes $\mathcal{V}_P$ in the training graph and labeled them as the target class $y_t$. Therefore, a GNN model trained on such poisoned graph will perform the following characteristics: **(i)** It will perform normally on the classification task of clean nodes. **(ii)** It will misclassify poisoned nodes with triggers as the target label. Specifically, any

node $v_p \in \mathcal{V}_P$ will be attached with an adaptive trigger $g_p = (A_p^g, X_p^g)$, and the poisoned node can be denoted as $\hat{v}_p = \mathcal{M}(v_p, g_p)$, where $\mathcal{M}(\cdot)$ stands for the attaching operation. The edge set of the connection between poisoned nodes and triggers and the internal connection relationship of triggers can be expressed as $\mathcal{E}_P$. Finally, the backdoored graph obtained by the attacker can be formulated as:

$$\tilde{\mathcal{G}}_T = \{\tilde{\mathcal{V}}_T, \tilde{\mathcal{E}}_T, \tilde{X}_T\}, \tag{1}$$

where $\tilde{\mathcal{V}}_T = \mathcal{V}_T \cup \tilde{\mathcal{V}}_P$, $\tilde{\mathcal{E}}_T = \mathcal{E}_T \cup \mathcal{E}_P$, and $\tilde{X}_T = X_T \cup X^g$.

**Defense Model.** Similar to the settings of most graph defense models, during the training phase, the defender can use the backdoor graph $\tilde{\mathcal{G}}_T$ to train a node classifier. However, the defender cannot obtain any knowledge about the set of poisoned nodes $\mathcal{V}_P$, the target label $y_t$, and the trigger structure $g_p$. While in the inference phase, the defender is provided with an unseen graph $\mathcal{G}_I$ for node classification. The overall defense problem can be defined as: Given a backdoored graph $\tilde{\mathcal{G}}_T$, we aim to train a defense model that is immune to backdoors, so that any node with added triggers can be protected from the impact of backdoors during the inference time while maintaining clean accuracy. The optimization problem can be formulated as:

$$\min_f \sum_{v_i \in \mathcal{V}_{I_c}} \mathcal{L}(f(v_i), y_i) + \sum_{v_p \in \mathcal{V}_{I_p}} \mathcal{L}(f(\hat{v}_p), y_t), \tag{2}$$

where $\mathcal{V}_{I_c}$ and $\mathcal{V}_{I_p}$ denote the clean node set and the poisoned node set in the inference graph, respectively. And $\mathcal{L}(\cdot)$ is the classification loss, such as the Cross-Entropy loss.

**Oracle Purification.** To provide a baseline upper bound on defense capabilities, we also constructed an oracle purification model. Specifically, it assumes that the defender has complete knowledge of the backdoor, including poisoned nodes, trigger structure, and target labels. It uses this *oracle data* to fine-tune the backdoor model, resulting in a model that represents an upper bound on the robustness of defense capabilities. Drawing on the definition of Exact Purification in Min et al. (2024), we also add some constraints specific to the unique structure of GNNs for fine-tuning. We use poisoned node set $\mathcal{V}_{\mathcal{P}}$ with clean label $y_i$, and clean node set $\mathcal{V}_{\mathcal{C}}$ for fine-tuning. For clean nodes, we still expect their classification to be as accurate as possible. While for poisoned nodes, we consider two aspects: firstly, we expect them to be classified correctly; secondly, to reduce the potential impact of the original backdoor, we hope that their predicted confidence in the target label is as small as possible. Therefore, we propose a comprehensive loss function:

$$\mathcal{L}_{op\_ft} = \mathcal{L}_{clean} + \lambda_{relearn} \cdot \mathcal{L}_{relearn} + \lambda_{suppress} \cdot \mathcal{L}_{suppress}. \tag{3}$$

## 4 ANALYZE THE VULNERABILITIES OF GRAPH BACKDOOR DEFENSES

### 4.1 PREPARING FOR RETUNING ATTACK

To verify the robustness of existing graph backdoor defense methods, we fine-tuned the defense model on poisoned data, similar to previous work. Specifically, we selected poisoned graphs with smaller poisoning ratios to fine-tune the defense model for a very small number of rounds. The smaller poisoning ratio was chosen to assume a more restricted poisoning environment. In other words, if a trained defense model performs poorly on such a dataset, its robustness is very poor. Furthermore, the selection of fewer rounds of fine-tuning is also to maintain a strict poisoning scenario. We hope to explore the true robustness of existing graph backdoor defense methods in this scenario.

### 4.2 TRAINING ORACLE PURIFICATION MODEL AS THE ROBUST UPPER BOUND

To ensure the rationality of our experiments, inspired by the work from Min et al. (2024), we also trained an idealized defense model with oracle knowledge as a control. In typical real-world scenarios, defenders often only receive information about the backdoor graph and the backdoor model, but lack knowledge of the distribution of poisoned nodes in the backdoor graph, the target label settings, and other information. However, in the context of an oracle purification model, the defender possesses absolute backdoor knowledge. Specifically, it explicitly knows the set of clean and poisoned nodes, as well as the target label. In this setting, the defender only needs to fine-tune the

backdoor model using backdoor samples with normal labels to significantly improve the robustness of the defense. Due to the unique structure of GNNs, which differs from DNNs, the purification schemes applicable to DNNs are ineffective on GNNs. This is because in GNN backdoor attacks, triggers are added to the target node in the form of subgraphs or single nodes. This then transmits malicious information to the target node during message passing and aggregation, thus enabling the backdoor attack. Furthermore, DNNs and GNNs employ completely different approaches during the purification phase. Backdoored DNNs use a small amount of clean data to fine-tune the backdoored model during the purification phase, hoping the model will forget the learned backdoor features and instead focus on more important benign features. Meanwhile, GNNs use unseen poisoned graphs rather than clean graphs during the purification phase. Therefore, GNN defenders typically employ algorithms to distinguish between potentially poisoned and clean nodes in the graph, and then modify the model's optimization objective, hoping to prioritize predictions for clean nodes while suppressing predictions for poisoned nodes.

Based on above reasons, we propose a new purification method to obtain the oracle model. We will precisely cleanse the backdoor model from three perspectives. First, a well-preserved cleansing model must maintain high classification accuracy on clean data. Therefore, we leverage the set of clean nodes, which is explicitly known to the defender, to optimize the model by reducing classification prediction loss. Second, a key difference from conventional defense models is that the Oracle defender precisely knows the set of poisoned nodes. Other defense methods typically use algorithms to analyze the set of potentially poisoned nodes. Inaccuracies in detecting poisoned nodes during this process can lead to a loss of defense performance. Therefore, we use the correct label class to guide the poisoned nodes as far away from the original target class as possible. Finally, to further suppress the adverse effects of the backdoor, we also aim to minimize the classification loss between the original poisoned nodes and the target label. We believe that purifying the backdoor model from these three perspectives is a more comprehensive, accurate, and precise cleaning solution suitable for GNNs.

### 4.3 EXPLORING ROBUST VULNERABILITIES OF DEFENSES

We apply RA attacks to a series of defense models and analyze how their attack success rates changed. In addition to the OP model, which represents the theoretical upper bound, we also selected a clean model as another baseline. This is because both the OP model and any other graph backdoor defense model are fine-tuned based on the backdoor model, meaning the model has previously learned information about the backdoor. This is completely different from a clean model, which is trained on completely clean data. When directly exposed to poisoned data during inference, its predictions should differ from those of any other defense models. The Figure 1 below shows the ASR changes of several models in response to UGBA attack on the Cora and Pubmed dataset.

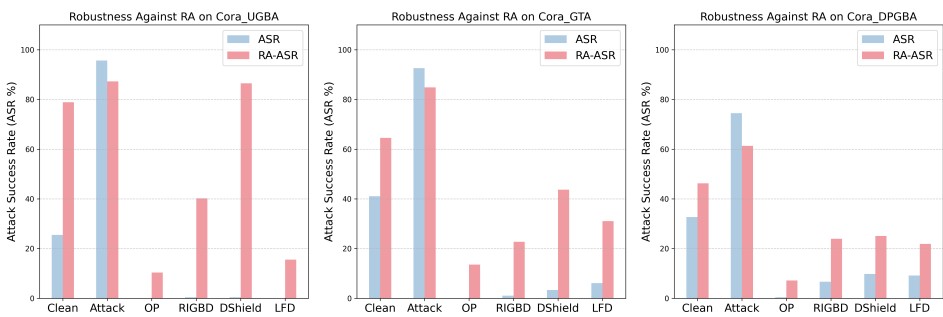

Figure 1: The ASR performance of defense methods against different attacks before and after RA.

The results show that the ASR of all defense models, including the clean model, has improved to varying degrees. The OP model, which represents the upper bound of defense capability, saw the smallest improvement in ASR. The other defense models all showed significant increases in ASR. Furthermore, the backdoor model's ASR decreased slightly, which is reasonable. Because we used poisoned data with a lower poison rate during the RA attack, the original backdoor model focused more on the feature representations of benign nodes and less on poisoned nodes. The clean model's

performance showed a similar trend to the defense model. This is because, although it had never seen poisoned data, when attacked by an RA, the target node, based on the key action of previously learned neighbor feature aggregation, incorporated the features of the malicious trigger into its own feature representation, resulting in a significant improvement in ASR.

# 5 LAYER-WISE FEATURE DIVERGENCE BASED ROBUST DEFENSE FRAMEWORK

## 5.1 HOW LLFC ANALYZES THE CONNECTIVITY OF THE MODEL ON GNN

LLFC represents the similarity between two types of output features at a certain level. Here, the comparison is between the output features of the interpolated model and the interpolated output features of the two models. A high similarity indicates a high degree of connectivity between the two models. This connectivity transforms optimization in the parameter space and optimization in the feature space; any deviation in the feature space will also lead to deviation in the parameter space. Extending this to backdoor attacks and defenses, if a cleansed model only learns to counteract backdoors at the final output level, but its internal feature extraction logic remains highly similar to that of the backdoor model, then this cleansing is fragile and superficially secure. Its connectivity with the original backdoor model at the parameter level remains strong. Attackers can easily activate backdoors lurking in similar feature spaces through small amounts of data fine-tuning (retraining attacks) or other adaptive attacks.

Based on the above analysis, we believe that directly adding constraints that destroy the connectivity of hierarchical features to the training objective can fundamentally change the internal operation mode of the model, thereby making the parameter space of the purified model deviate from the backdoor model to a greater extent.

## 5.2 POISONED NODES DETECTION BASED ON FEATURE PERTURBATION

Defense that only optimizes the objective function is incomplete. Similar to other defense methods, we also propose a new poisoned node selection strategy that uses random feature perturbations to distinguish poisoned nodes from clean nodes. Specifically, the prediction of poisoned nodes is extremely dependent on triggers that are simple in structure but have a strong impact. Then, the prediction results of poisoned nodes should be insensitive to the feature perturbations of their benign neighbors because they mainly receive strong information from triggers. On the contrary, the prediction results of a clean node are an aggregation of the features of its benign neighbors, so changes in neighbor information will have a greater impact on its prediction results. Taking advantage of this difference in influence, we add a random noise $\epsilon$ sampled from normal distribution $\mathcal{N}(\mu, \sigma)$ to each node $v_i \in \mathcal{D}_T$ in poisoned graph. Then, we use the feature $x_i + \epsilon$ of node $v_i$ to get its prediction logits. Based on the gap between the original logits, we sort all nodes in ascending order of the gap size and get the item sequence $\mathcal{S}$. We use the prediction label of the first item in $\mathcal{S}$ as target label $y_p = y_{\mathcal{S}_{(1)}}$. The final candidate poisoned node set is defined as:

$$\mathcal{P} = \mathcal{S}_{[:k]}, \quad k = \min\{t | y_{\mathcal{S}_{(t)}} \neq y_{\mathcal{S}_{(1)}} \quad \text{and} \quad y_{\mathcal{S}_{(t+1)}} \neq y_{\mathcal{S}_{(1)}}\}. \tag{4}$$

Then, we set the rest nodes in $\mathcal{D}_T$ as clean nodes set $\mathcal{C}$. They will be used to optimize different objective loss terms in the subsequent fine-tuning stage.

## 5.3 OVERALL IMPLEMENTATION OF THE LFD ROBUST FRAMEWORK

In the overall optimization goal, we mainly consider three aspects. Firstly, for node $v_i \in \mathcal{C}$, we want it to be classified as accurately as possible. Therefore, we use the common cross-entropy loss to calculate the loss between them and denote as $\mathcal{L}_{clean}$. Secondly, for node $v_j \in \mathcal{P}$, we propose to minimize its prediction confidence on the target class $y_p$ and denote as $\mathcal{L}_{suppress}$. Thirdly, the loss item based on LLFC is denoted as $\mathcal{L}_{llfc}$. Specifically, we denote the weight parameter of the backdoor model as $W_0$, and the current defense model as $W_1$. Then, we construct the weight parameters of the interpolation model as:

$$W_t = (1 - t) \cdot W_0 + t \cdot W1, \tag{5}$$

here, $t \in [0, 1]$ is an interpolation coefficient. Then, we use $W_0$, $W_1$ and $W_t$ to get the feature map of each layer of the victim model. The corresponding feature representation can be denoted as: $h_l(x; W_0)$, $h_l(x; W_1)$ and $h_l(x; W_t)$ for layer $l$. According to the definition of LLFC, if $W_0$ and $W_1$ are linear mode connected, the expected feature map is:

$$h_{exp} = (1 - t) \cdot h_l(x; W_0) + t \cdot h_l(x; W_1), \tag{6}$$

here, $h(\cdot)$ represents the feature function of the victim model. But our goal is to expect the weight parameter space of the defense model to be as far away from the backdoor model as possible, that is, the worse the linear mode connectivity between the two, the better. Therefore, we define the third part of the loss function as:

$$\mathcal{L}_{llfc} = \sum_l \mathcal{L}_{llfc_l} = \sum_l \text{cosine}(h_l(x; W_t), h_{exp}). \tag{7}$$

In conclusion, the final complete loss function is expressed as:

$$\mathcal{L}_{total} = \mathcal{L}_{clean} + \lambda_1 \cdot \mathcal{L}_{suppress} + \lambda_2 \cdot \mathcal{L}_{llfc} \tag{8}$$

# 6 EXPERIMENTS

In this section, we conduct several experiments to answer the following research questions: **(RQ1)** How effective is LFD in the robust defense performance before and after retuning attack? **(RQ2)** How does LFD perform in terms of layer-wise feature connectivity? **(RQ3)** How do different loss terms of the optimization objective affect the performance of the LFD model?

## 6.1 EXPERIMENTAL SETUP

**Datasets.** We conduct experiments on benchmark datasets that are widely used in node classification tasks, i.e., Cora, PubMed (Sen et al., 2008), and OGB-arxiv (Hu et al., 2020). More details about the datasets are summarized in Appendix A.1.

**Attack Methods & Attack settings.** In order to compare the defense performance of different defense methods, we evaluate on three state-of-the-art graph backdoor attack methods, UGBA (Dai et al., 2023), GTA (Xi et al., 2021) and DPGBA (Zhang et al., 2024). More details about the attack methods are summarized in Appendix A.2.

**Defense Methods.** To demonstrate the performance of our proposed LFD training strategy, we also conduct experiments on representative graph backdoor defense methods. We choose RIGBD (Zhang et al., 2025) and DShield (Yu et al., 2025) for comparison. More details about the defense methods are summarized in Appendix A.3.

## 6.2 PERFORMANCE OF DIFFERENT DEFENSE METHODS ON GENERAL CAPABILITIES

To answer **RQ1**, we conduct a series of experiments on three common datasets and three backdoor attack methods. The experimental results are shown in Table 1. The results show that our LFD performs best on most datasets and attack methods. Specifically, LFD exhibits lower RA-ASR while maintaining comparable CA and ASR performance. We conducted all experiments three times and reported the final average results.

## 6.3 PERFORMANCE OF LAYER FEATURE DIVERGENCE

To answer **RQ2**, we analyze the distance results between the layer-by-layer output features of the victim model, and the results are shown in Figure 2. The results in the figure show that, overall, the blue bar in each figure is significantly higher than the bars of other colors. This is because the blue bar measures the absolute distance between the defense model and the backdoor model in the output feature space of that layer. A larger blue bar indicates a greater difference between the learned features and the backdoor features, and a greater distance from the backdoor model. It can be seen that the OP model, which serves as the theoretical bound, has the highest blue bar, and our LFD also has a higher bar than other defense methods. Furthermore, the blue bar decreases

Table 1: Performance of different defense methods on metrics CA, ASR and RA-ASR.

| Defense | Dataset | UGBA | | | GTA | | | DPGBA | | |
|---|---|---|---|---|---|---|---|---|---|---|
| | | CA ↑ | ASR ↓ | RA-ASR ↓ | CA ↑ | ASR ↓ | RA-ASR ↓ | CA ↑ | ASR ↓ | RA-ASR ↓ |
| Clean Model | | 83.33 | 25.50 | 78.88 | 82.96 | 41.04 | 64.54 | 82.96 | 32.67 | 46.22 |
| No Defense | | 81.48 | 95.62 | 87.25 | 83.70 | 92.62 | 84.86 | 84.81 | 74.50 | 61.35 |
| OP Model | Cora | 81.11 | 0.00 | 10.36 | 81.85 | 0.00 | 13.55 | 81.11 | 0.40 | 7.17 |
| RIGBD | | 81.85 | 0.40 | 40.42 | 80.37 | 1.07 | **22.75** | 75.56 | 6.64 | 23.94 |
| DShield | | 82.22 | 0.40 | 86.45 | 80.37 | 3.32 | 43.68 | 80.74 | 9.81 | 25.10 |
| LFD(Ours) | | 83.70 | 0.00 | **15.54** | 83.70 | 6.10 | 31.07 | 83.33 | 9.16 | **21.91** |
| Clean Model | | 85.49 | 68.04 | 59.82 | 84.42 | 100.00 | 98.91 | 85.03 | 80.54 | 79.87 |
| No Defense | | 85.64 | 85.32 | 82.97 | 85.03 | 98.57 | 97.82 | 85.03 | 79.70 | 81.71 |
| OP Model | PubMed | 82.09 | 2.27 | 7.72 | 80.42 | 0.25 | 1.59 | 52.51 | 0.00 | 0.80 |
| RIGBD | | 83.51 | 2.14 | 22.57 | 84.32 | 3.50 | 55.13 | 84.47 | 3.39 | 8.33 |
| DShield | | 83.87 | 2.97 | 86.41 | 83.61 | 0.80 | **1.83** | 83.51 | 3.37 | 8.07 |
| LFD(Ours) | | 84.88 | 0.67 | **11.11** | 85.03 | 0.80 | 1.90 | 85.13 | 8.78 | **7.65** |
| Clean Model | | 60.41 | 11.42 | 20.69 | 60.38 | 0.84 | 13.58 | 60.36 | 8.72 | 13.80 |
| No Defense | | 59.32 | 99.79 | 79.55 | 60.07 | 94.85 | 85.44 | 59.77 | 93.60 | 84.53 |
| OP Model | OGB-arxiv | 52.88 | 0.97 | 1.02 | 58.44 | 0.00 | 2.43 | 58.60 | 1.07 | 5.48 |
| RIGBD | | 60.43 | 1.02 | 27.28 | 60.66 | 7.23 | 17.19 | 61.04 | 5.10 | 20.34 |
| DShield | | 58.95 | 3.09 | 11.18 | 57.71 | 6.58 | 24.04 | 58.81 | 2.42 | 16.89 |
| LFD(Ours) | | 59.94 | 0.40 | **4.80** | 58.89 | 6.29 | **12.87** | 59.22 | 3.61 | **9.86** |

in each sub-figure at the second layer. This is because the defense models share a common task trend: ultimately, they all achieve high CA and low ASR, resulting in a similar convergence at the second layer. The three colored bars consistently show low values, rising slightly in the second layer. Due to the computational method of the first layer of GCN, and the sole nonlinear factor, the ReLU function, also discovered in the paper Zhou et al. (2023), when two models are in similar loss basins, the ReLU function behaves very close to linearly. All defense methods are fine-tuned based on the backdoor model, meaning they have the same starting point and therefore occupy the same large loss basin. In the second layer of GCN, due to the nonlinear activation of the first layer, the relationship is not completely linear, and the bar will rise slightly.

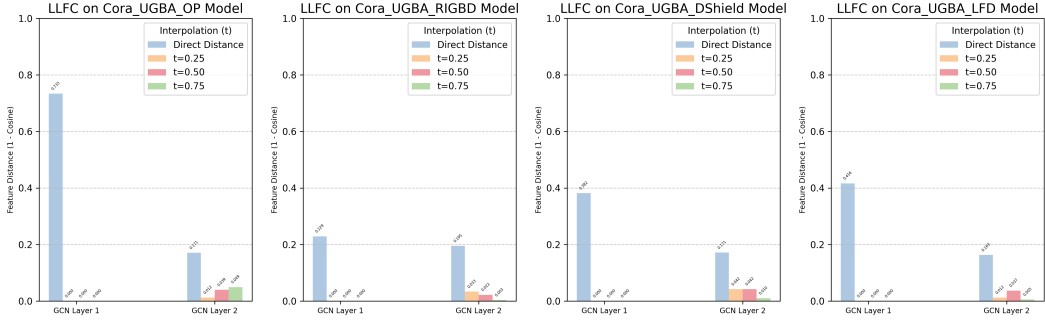

Figure 2: Performance of defense methods on output feature distance of a two-layer GCN.

Furthermore, we conducted a series of ablation experiments to analyze the impact of applying divergence optimization to different layers. Specifically, in the standard LFD method, we chose the average feature similarity of the outputs of two GCN layers as the optimization term for the loss. To verify the impact of different layers, we selected only the output features of one layer as an optimization term and analyzed the actual effect of applying divergence optimization to different layers based on three metrics: CA, ASR and RA-ASR. The results are shown in the Table 2 below, showing that using the average feature of two layers has a better overall effect. In addition, we also found that feature divergence works better in the first layer than in the second layer. This is because for models like GCN with very few layers, the first layer is usually used for original feature extraction and is far from backdoor features. Therefore, applying divergence optimization here can effectively keep the defense model away from backdoors. The output of the second layer is generally used to achieve the task objective, so the effect of divergence is generally less significant.

Table 2: Ablation study on different divergence layers

| Dataset | Layer1 | Layer2 | UGBA | | | GTA | | |
|---------|--------|--------|------|-----|--------|------|-----|--------|
| | | | CA | ASR | RA-ASR | CA | ASR | RA-ASR |
| Cora | ✓ | | 81.11 | 7.99 | 20.32 | 82.22 | 20.26 | 57.71 |
| | | ✓ | 82.12 | 12.51 | 27.49 | 81.11 | 31.65 | 74.90 |
| | ✓ | ✓ | **83.70** | **0.00** | **15.54** | **83.70** | **6.10** | **31.07** |
| PubMed | ✓ | | 83.41 | 5.74 | 17.05 | 85.74 | 2.67 | 16.54 |
| | | ✓ | 83.26 | 4.19 | 18.64 | 85.34 | 3.11 | 29.76 |
| | ✓ | ✓ | **84.88** | **0.67** | **11.11** | **85.03** | **0.80** | **1.90** |
| OGB-arxiv | ✓ | | 60.23 | 2.97 | 6.64 | 60.25 | 19.75 | 27.67 |
| | | ✓ | 60.93 | 3.11 | 6.73 | 60.69 | 20.69 | 27.45 |
| | ✓ | ✓ | **59.94** | **0.40** | **4.80** | **58.89** | **6.29** | **12.87** |

To visually demonstrate the defense performance of different methods, we selected the t-SNE results of the last layer output space of the Backdoor model, OP model, RIGBD, DShield and LFD for visualization. The results are shown in Figure 3 and Figure 4 below. As can be seen in the figures, for the Backdoor model, poisoned nodes (black) are mostly clustered in the target class (orange) cluster, indicating that the attack model performs well. OP model, as the theoretical bound of defense, possesses complete knowledge of the backdoor, thus accurately mapping poisoned nodes back to their original classes. RIGBD is relatively messy overall, without forming clear classification boundaries, indicating that the backdoor may be easily reactivated. Although DShield maps some nodes back to their original classes, it mainly ignores the influence of triggers through semantic and attribute differences, without fundamentally severing the mapping from triggers to target labels; therefore, some poisoned nodes are still very close to the target class cluster. LFD method utilizes the principle of hierarchical feature divergence to forcibly isolate poisoned nodes into a cluster as far away from the target label as possible, creating a greater security margin for the defense model.

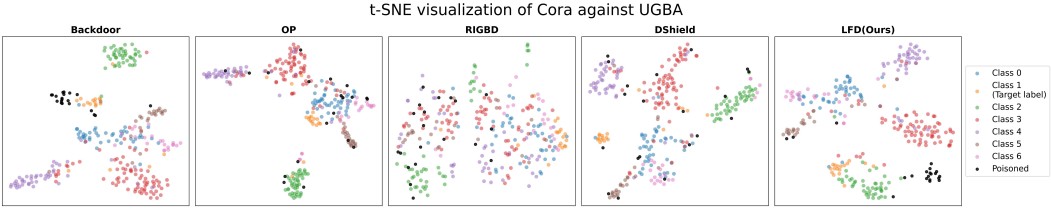

Figure 3: T-SNE visualization result of different models on Cora against UGBA.

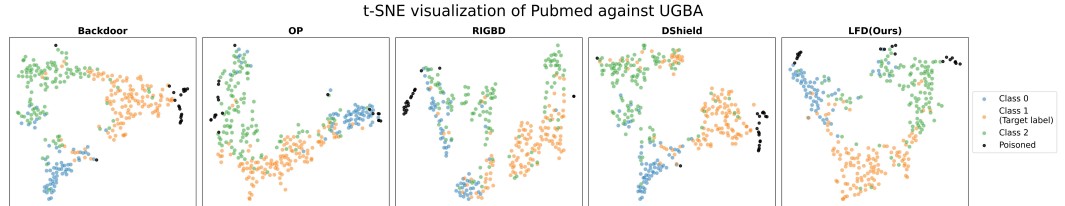

Figure 4: T-SNE visualization result of different models on PubMed against UGBA.

## 6.4 ABLATION STUDY AND HYPERPARAMETER ANALYSIS

To answer **RQ3**, we explore the contributions of the three components of the objective function used in training the **LFD** model. Table 3 shows the ablation comparison results of the attack method UGBA on the Cora dataset. As we have analyzed above, we can see that the three losses each have their own optimization objectives and achieve good results. Specifically, the clean loss, as the dominant loss term for clean node classification task, performs well on the CA metric, but it focuses almost exclusively on clean node features, resulting in high ASR and RA-ASR. The suppress loss primarily aims to reduce the prediction confidence of the poisoned node on the target label, hoping that the model will forget as much as possible the backdoor knowledge learned earlier. Therefore,

it performs very well on the ASR metric. However, it only optimizes distance from the target label, and the model parameters do not move away from the original backdoor model. Therefore, after RA attack, the ASR rises again to a large value. The llfc loss, on the other hand, focuses on the distance between inter-layer features, and its main goal is to push the defense model as far away from the original backdoor model as possible. However, the loss basin of the backdoor model is likely to be a flat, wide area. Although it moves away from the starting point, it is likely that only the horizontal distance increases, resulting in a high ASR. However, since it controls the model to move away from the original backdoor as much as possible, its ASR does not increase much after the RA attack. By considering three loss terms simultaneously, we not only ensure the classification accuracy on clean nodes, but also guide the defense model to update in a direction that is far away from the backdoor basin in both horizontal and vertical distances, thereby finding a more robust solution.

Table 3: The ablation study of the loss components for LFD model.

| Dataset | clean_loss | suppress_loss | llfc_loss | CA | ASR | RA-ASR |
|---|---|---|---|---|---|---|
| Cora | ✓ | | | 82.59 | 82.47 | 88.45 |
| | ✓ | ✓ | | 83.33 | 0.00 | 70.92 |
| | ✓ | | ✓ | **85.19** | 62.95 | 66.53 |
| | ✓ | ✓ | ✓ | 83.70 | **0.00** | **15.54** |

We also conducted a detailed hyperparameter analysis on the weights of the loss terms. The weights of the three loss terms in the loss function are denoted as $\mathcal{L}_{clean}$, $\mathcal{L}_{suppress}$ and $\mathcal{L}_{llfc}$, respectively. We mainly analyze the influence of the weights of the latter two parameters. Specifically, we fixed one parameter and varied the other across [0.1, 1.0, 10.0] to observe the impact on CA, ASR and RA-ASR. We provided the visualization result of OGB-arxiv against GTA in Figure 5 below. The results show that as analyzed above, there is a trade-off between the two loss terms.

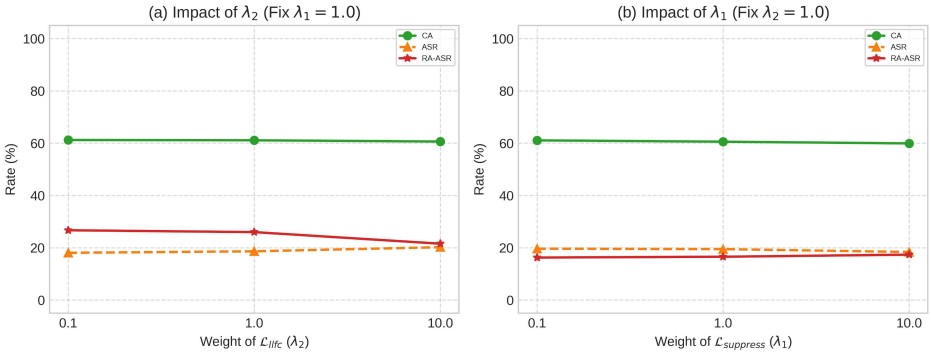

Figure 5: Hyperparameter analysis of optimization terms of LFD on OGB-arxiv against GTA.

## 7 CONCLUSION

In this paper, we first systematically analyze the purification robustness of existing graph backdoor defense methods. We find that while existing defense models exhibit low attack success rates, their ASR performance can be significantly improved with only a few epochs of fine-tuning using a small number of poisoned samples. To further explore this phenomenon, we utilize the concept of layer-wise linear feature connectivity to conduct a detailed analysis of the internal feature space of the defense model. The results show that existing defense methods lack a clear separation between the output features of each layer within the model and the backdoor model, resulting in residual backdoor features being easily reactivated. Inspired by this, we propose a more robust defense strategy, called hierarchical feature divergence (LFD), which explicitly distances the features between the defense model and the backdoor model as much as possible during training, thereby improving the robustness of the defense. Extensive experiments demonstrate that our approach improves the robustness of the defense model while maintaining a high clean accuracy.

## 8 ETHICS STATEMENT

In this paper, we propose a robust graph backdoor defense method based on linear feature divergence. This technique itself should not pose serious ethical issues.

## 9 REPRODUCIBILITY STATEMENT

The experimental and implementation details can be found in Section 6. The code of the key part is now publicly available at: `https://anonymous.4open.science/r/LFD-main-D8FD`, and the complete code will be uploaded later.

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

# A  MORE DETAILS ABOUT EXPERIMENTAL SETUP

## A.1  DATASETS

We selected several datasets that are most commonly used for GNN node classification tasks for experiments. **Cora** dataset consists of academic papers in the field of computer science. Each node in the network represents a paper, and if a paper cites another paper, there is an edge between them. Its task is to predict the academic field (category) of each paper based on the paper's word vectors (features) and citation relationships (graph structure). **PubMed** is a database containing a large number of scientific publications on diabetes. Nodes represent papers, and edges represent citation relationships. **OGB-arxiv** is an academic paper citation network dataset containing citation relationships of hundreds of millions of computer science papers, and is commonly used for node prediction tasks. The detailed information about the dataset is shown in the Table 4 below.

Table 4: The statistics of datasets.

| Dataset | # Nodes | # Edges | # Features | # Classes |
|---------|---------|---------|------------|-----------|
| Cora | 2,708 | 5,429 | 1,443 | 7 |
| PubMed | 19,717 | 44,338 | 500 | 3 |
| OGB-arxiv | 169,343 | 1,166,243 | 128 | 40 |

## A.2  ATTACK METHODS & ATTACK SETTINGS

We select three common backdoor attack methods to train the victim model.

- **UGBA.** It proposes an unnoticeable backdoor attack method under a limited attack budget, consciously selects more important nodes to inject triggers during the poisoning phase, and adaptively generates triggers for each node.

- **GTA.** Graph Trojaning Attack (GTA) is the first work of backdoor attack on graph area. It proposes a technique for generating adaptive triggers, dynamically adjusting the trigger features for each input without the need for downstream models or any fine-tuning strategies. In addition, it can be extended to transductive node classification tasks and inductive graph classification tasks.

- **DPGBA.** In order to solve the out-of-distribution problem of existing self-supervised backdoor attacks, a new distribution-preserving attack is proposed. By reducing the distribution distance between poisoned samples and clean samples, the poisoned samples are converted into in-distribution data. At the same time, the distribution of poisoned data in the target class distribution is expanded to a wider area, alleviating the concentration problem.

## A.3  DEFENSE METHODS

We select two graph backdoor defense methods that currently perform SOTA for comparison.

- **RIGBD.** It uses a random edge-dropping strategy to detect backdoors and identify specific poisoned nodes. It also theoretically verifies that this edge-dropping method can effectively distinguish poisoned nodes from clean nodes. It also introduces a new robust training strategy that can offset the influence of triggers.

- **DShield.** It proposes a differential learning-based approach to defend against backdoor triggers, capable of defending against both dirty-label and clean-label backdoor attacks. The approach improves the model's defensive performance through three modules: auxiliary model training, differential matrix construction, and backdoor filtering and retraining.

# B  ADDITIONAL EXPERIMENTAL RESULTS OF LAYER FEATURE DIVERGENCE

We also perform an ablation analysis on the optimization function of the OP model. We compared the clean accuracy, original attack success rate, and post-RA attack success rate of the OP model

Table 5: The ablation study of the loss components for OP model.

| Dataset | clean_loss | relearn_loss | suppress_loss | CA | ASR | RA-ASR |
|---------|:---:|:---:|:---:|:---:|:---:|:---:|
| Cora | ✓ | | | **85.93** | 74.10 | 76.10 |
| | ✓ | ✓ | | 83.33 | 41.43 | 53.78 |
| | ✓ | | ✓ | 85.19 | 0.00 | 27.09 |
| | ✓ | ✓ | ✓ | 81.11 | **0.00** | **10.36** |

under different loss function optimizations on the Cora dataset. The results are shown in the Table 5 below. Similarly, optimizing the OP model using three loss terms can yield the best performance.

In addition, we further analyzed the performance of different defense methods in terms of inter-layer feature similarity distance on different datasets and attacks. As shown in Figure 6, the OP model has the highest direct feature distance which means a better defense capability. In the first layer output, the DShield method outperforms our LFD method. However, LFD exhibits a larger feature distance, i.e., smaller feature similarity, in the second layer output. This indicates that LFD performs better in terms of features far from the backdoor.

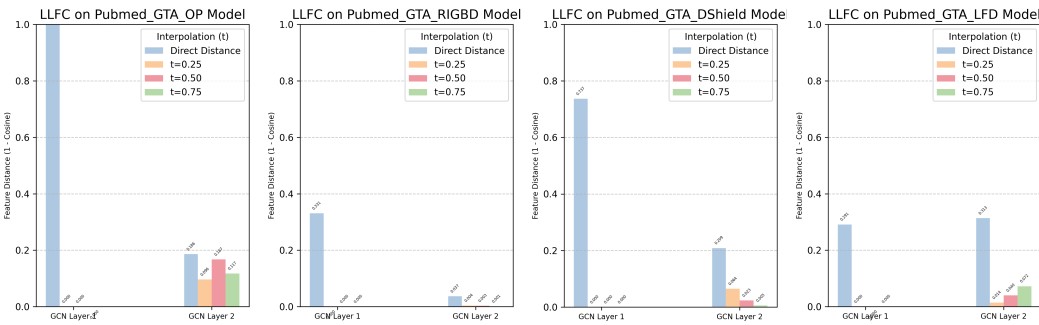

Figure 6: The performance of methods before and after RA.

## C ADDITIONAL EXPERIMENTAL RESULTS OF THE CORRELATION BETWEEN LLFC AND RA-ASR

Quantitative analysis of LLFC distance and RA-ASR provides a more intuitive understanding of the crucial role of increasing hierarchical feature distance in improving the robustness of the defense model. Therefore, we use the following visualization to illustrate this. As shown in the Figure 7, the graph illustrates the approximate relationship between LLFC and RA-ASR for different defense methods. Taking any subgraph as an example, the dots to the right of the horizontal line represent the LLFC distance of L1, and the crosses to the left represent the LLFC distance of L2. The horizontal axis represents LLFC distance, and the vertical axis represents RA-ASR. It can be seen that OP, as the theoretical upper bound of the defense model, has the largest overall LLFC. Simultaneously, RA-ASR is the smallest. Compared to the other two defense methods, our LFD method has a larger LLFC and a smaller RA-ASR. Overall, a roughly inverse relationship between LLFC and RA-ASR can also be observed.

## D ADDITIONAL EXPERIMENTAL RESULTS OF HYPERPARAMETER ANALYSIS

We also conducted further hyperparameter analysis on different datasets and attack methods, and the results are shown in the Figure 8. The figure shows the hyperparameter analysis results of the two loss terms when the LFD method is attacked by the UGBA backdoor on the Cora dataset. Similarly, there is a trade-off in the performance of the two loss terms, $\mathcal{L}_{suppress}$ and $\mathcal{L}_{llfc}$, on ASR and RA-ASR, while their overall performance on CA is relatively stable.

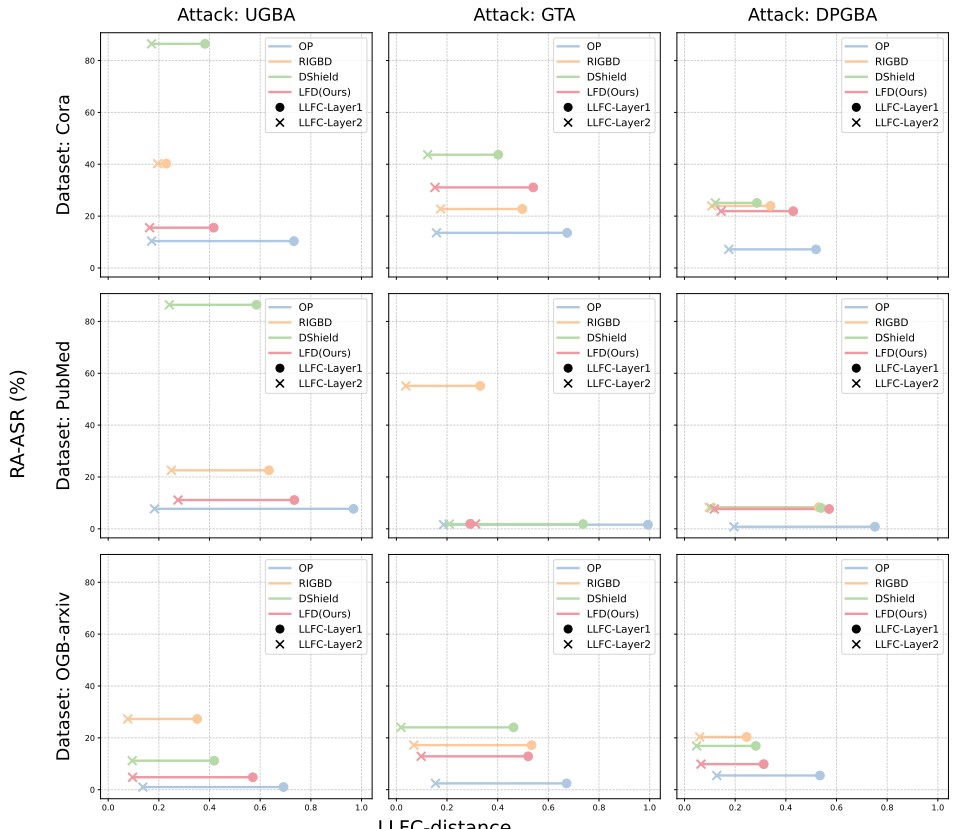

Figure 7: Visual analysis of the quantitative relationship between LLFC and RA-ASR.

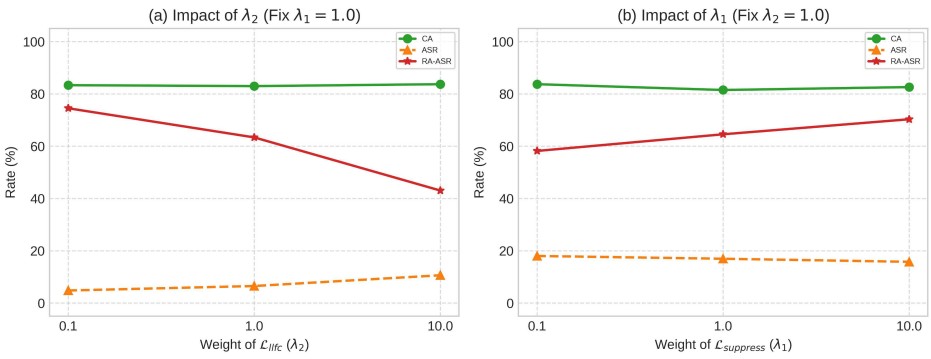

Figure 8: Hyperparameter analysis of optimization terms of LFD on Cora against UGBA.

# E  ADDITIONAL EXPERIMENTAL RESULTS OF THE POISONED NODE DETECTION ALGORITHM

Our poisoned node identification algorithm primarily distinguishes between poisoned and clean nodes by leveraging the varying impacts of random perturbations in neighbor features on prediction results. For clean nodes, predictions are highly dependent on the feature aggregation of benign neighbors; therefore, changes in neighboring features significantly affect predictions. For poisoned nodes, predictions are highly dependent on triggers, and the mapping from triggers to target labels is

typically overfitted. Therefore, even adding perturbations to neighbor triggers does not significantly alter prediction results. We emphasize that the purpose of feature perturbations is not to simulate triggers from different attack methods (such as structural perturbations), but rather to identify nodes insensitive to perturbations in neighbor features. To visually demonstrate the effectiveness of the poisoned node identification algorithm, we supplemented the following experiments. The overall defense mode of the graph backdoor defense method RIGBD is the same as our LFD, primarily focusing on poisoned node identification in the first stage. Therefore, we selected three metrics—recall, precision, and F1-score—to compare the recognition performance of the two methods. The results in the Table 6 below show that although our recognition algorithm is slightly weaker than RIGBD, LFD's defense robustness is far superior to RIGBD in the second stage of defense training. This further demonstrates the crucial role of feature divergence optimization training.

Table 6: Performance of poisoned node detection algorithms.

| Datasets | Attacks | RIGBD | | | | | LFD | | | | |
|---|---|---|---|---|---|---|---|---|---|---|---|
| | | Recall | Precision | F1-score | ASR | RA-ASR | Recall | Precision | F1-score | ASR | RA-ASR |
| Cora | UGBA | 0.968 | 1.000 | 0.984 | 0.40 | 40.24 | 0.900 | 1.000 | 0.947 | 0.00 | **15.54** |
| | GTA | 0.912 | 0.940 | 0.926 | 1.07 | **22.75** | 0.875 | 1.000 | 0.933 | 6.10 | 31.07 |
| | DPGBA | 1.000 | 1.000 | 1.000 | 6.64 | 23.94 | 0.869 | 0.914 | 0.891 | 9.16 | **21.91** |
| OGB-arxiv | UGBA | 0.956 | 0.963 | 0.959 | 1.02 | 27.28 | 0.852 | 0.910 | 0.880 | 0.40 | **4.80** |
| | GTA | 0.849 | 0.903 | 0.875 | 7.23 | 17.19 | 0.811 | 0.858 | 0.834 | 6.29 | **12.87** |
| | DPGBA | 0.905 | 0.939 | 0.922 | 5.10 | 20.34 | 0.862 | 0.851 | 0.856 | 3.61 | **9.86** |
| PubMed | UGBA | 0.887 | 0.966 | 0.925 | 2.14 | 22.57 | 0.812 | 0.957 | 0.879 | 0.67 | **11.11** |
| | GTA | 0.820 | 0.905 | 0.860 | 3.50 | 55.13 | 1.000 | 0.976 | 0.988 | 0.80 | **1.90** |
| | DPGBA | 0.850 | 0.910 | 0.879 | 3.39 | 8.33 | 0.785 | 0.850 | 0.816 | 8.78 | **7.65** |

# F ADDITIONAL EXPERIMENTAL RESULTS OF THE COMPUTATIONAL COSTS OF LFD

We also analyzed the training and testing times of different defense methods, as well as the maximum memory usage during training. The results are shown in the Table 7 below. Although our LFD method outperforms the standard model and RIGBD in terms of training time and memory usage, these costs can be considered one-time expenses in the defense process. Once LFD is trained, its computational cost during the inference phase is close to that of the standard GCN model and lower than other defense methods.

Table 7: Computational cost analysis (time and memory)

| Defense | Dataset | Atk-UGBA | | Atk-GTA | | Atk-DPGBA | |
|---|---|---|---|---|---|---|---|
| | | Train/Test_time | Memory | Train/Test_time | Memory | Train/Test_time | Memory |
| Clean Model | | 0.97s/0.65s | 0.0378GB | 1.00s/0.64s | 0.0452GB | 1.03s/0.65s | 0.0377GB |
| RIGBD | Cora | 4.00s/1.86s | 0.3295GB | 5.96s/1.88s | 0.3427GB | 3.53s/1.04s | 0.3581GB |
| DShield | | 108.00s/2.80s | 1.3794GB | 115.51s/3.81s | 1.6733GB | 67.65s/2.76s | 1.5937GB |
| LFD(Ours) | | 9.8s/0.79s | 0.3454GB | 7.14s/0.62s | 0.3825GB | 5.69s/0.79s | 0.4671GB |
| Clean Model | | 3.04s/1.86s | 0.1061GB | 3.13s/1.84s | 0.1154GB | 3.27s/1.80s | 0.1306GB |
| RIGBD | PubMed | 14.47s/2.55s | 0.4616GB | 16.35s/3.66s | 0.4822GB | 17.46s/3.14s | 0.5186GB |
| DShield | | 126.84s/5.04s | 1.5477GB | 126.48s/5.75s | 1.6943GB | 116.45s/5.50s | 1.6854GB |
| LFD(Ours) | | 19.56s/2.06s | 0.4983GB | 19.52s/2.97s | 0.5559GB | 19.30s/2.22s | 0.5560GB |
| Clean Model | | 15.13s/51.46s | 1.0222GB | 15.12s/64.36s | 1.0247GB | 15.12s/61.43s | 1.0231GB |
| RIGBD | OGB-arxiv | 21.13s/74.40s | 2.8764GB | 21.37s/71.48s | 2.7842GB | 20.23s/73.76s | 2.7628GB |
| DShield | | 920.39s/85.19s | 4.9430GB | 884.26s/96.16s | 4.9484GB | 953.05s/117.26s | 4.9362GB |
| LFD(Ours) | | 25.76s/61.62s | 2.9437GB | 32.74s/67.22s | 2.9792GB | 24.17s/63.74s | 2.8241GB |

# G ADDITIONAL EXPERIMENTAL RESULTS OF CLEAN-LABEL SETTINGS

Our proposed poisoned node identification algorithm based on feature perturbation is currently only applicable to dirty-labeled nodes. This is because it uses the variance of the prediction results caused by feature perturbation to distinguish between poisoned and clean nodes. However, in the clean-label setting, poisoned nodes and the target class share the same label, thus limiting the algorithm's

effectiveness. But our LFD can generalize to the clean-label setting using a second-stage feature divergence training based on LLFC. Specifically, the first stage aims to identify potential poisoned and clean nodes from the given data. Once the corresponding sets of nodes of different categories are obtained, a more robust defense model can be trained using the divergence principle of the second stage. Therefore, the first-stage poisoned node identification algorithm is actually replaceable. To verify the effectiveness of our argument, we selected poisoned node sets identified by the RIGBD and DShield methods and trained the defense on the LFD method. The experimental results are shown in the Table **??** below. It can be seen that the defense using LFD is more robust than the defenses of the original methods.

Table 8: Performance of layer feature divergence training under the dirty-label setting.

| Dataset | Defense | UGBA | | | GTA | | |
|---------|---------|------|-----|--------|-----|-----|--------|
| | | CA | ASR | RA-ASR | CA | ASR | RA-ASR |
| Cora | RIGBD | 81.85 | 0.40 | 40.24 | 80.37 | 1.07 | 22.75 |
| | RIGBD-LLFC (Dirty) | 83.81 | 0.30 | **31.08** | 81.11 | 1.82 | **17.15** |
| | DShield | 82.22 | 0.40 | 86.45 | 80.37 | 3.32 | 43.68 |
| | DShield-LLFC (Dirty) | 83.12 | 0.00 | **44.62** | 80.37 | 3.11 | **33.07** |
| PubMed | RIGBD | 83.51 | 2.14 | 22.57 | 84.32 | 3.50 | 55.13 |
| | RIGBD-LLFC (Dirty) | 83.74 | 3.27 | **10.21** | 83.29 | 3.66 | **49.10** |
| | DShield | 83.87 | 2.97 | 86.41 | 83.61 | 0.80 | **1.83** |
| | DShield-LLFC (Dirty) | 83.54 | 2.56 | **28.14** | 83.54 | 0.80 | 1.84 |

Furthermore, the DShield method can also be implemented with a clean-label setting. Therefore, we also conducted related experiments using the **GCBA** clean-label attack as shown in Table 9. It can be seen that for clean-label backdoor attacks, LFD training also outperforms its original performance. In conclusion, although our method cannot be directly used with a clean-label setting, it can serve as a general training paradigm, directly applied after various poisoned node identification algorithms to further improve the robustness of the defense model.

Table 9: Performance of layer feature divergence training under the clean-label setting.

| Dataset | Defense | UGBA | | |
|---------|---------|------|-----|--------|
| | | CA | ASR | RA-ASR |
| Cora | DShield | 82.59 | 2.76 | 52.11 |
| | DShield-LLFC (Clean) | 81.89 | 2.81 | **15.54** |
| PubMed | DShield | 84.23 | 3.07 | 37.66 |
| | DShield-LLFC (Clean) | 83.96 | 3.04 | **19.63** |

## H   LIMITATIONS AND FUTURE WORK

While our method can be used as a training paradigm in clean-label attack scenarios, its current limitations in the first-stage poisoned node detection algorithm are indeed due to the dirty label setting. In future work, we will explore poisoned node detection algorithms specifically for clean-label attacks. Furthermore, we will attempt to extend our work to larger-scale models.

## I   DECLARATION OF LLM USAGE

This paper utilizes LLMs to polish sentences and correct spellings.

