# OpenReview forum: "On the Fragility of Graph Backdoor Defenses: A Robust Strategy via Layer-wise Feature Divergence"
_ICLR.cc/2026/Conference — Submitted to ICLR 2026_

### Official Review · Reviewer_vCbK · 2025-10-31

**Soundness:** 3
**Presentation:** 3
**Contribution:** 3
**Rating:** 6
**Confidence:** 3

**Summary:**

This paper studies the robustness of graph neural network (GNN) backdoor defenses, which aim to remove or mitigate malicious triggers embedded during training. The authors find that existing defenses, though seemingly effective, are fragile—their attack success rate (ASR) can quickly rebound when the model is fine-tuned even slightly with poisoned data. To explain this, the authors analyze feature-space similarity between purified and backdoored models using Layer-wise Linear Feature Connectivity (LLFC), revealing persistent feature-level correlations. Based on this insight, they propose a novel defense strategy called Layer-wise Feature Divergence (LFD), which explicitly maximizes divergence in layer-wise feature representations between the defense model and the original backdoored model. Experiments on standard datasets (Cora, PubMed, OGB-Arxiv) and several attack settings (UGBA, GTA, DPGBA) show that LFD achieves improved robustness (low ASR and RA-ASR) while maintaining high clean accuracy.

**Strengths:**

1. Novel and well-motivated problem formulation.
The authors are the first to systematically evaluate the robustness of GNN backdoor defenses beyond one-shot purification. The observation that existing defenses are easily “reinfected” by minimal fine-tuning is original and practically important.

2. Insightful analysis via LLFC.
Using LLFC to quantify feature-space similarity provides a fine-grained, interpretable diagnostic for understanding residual backdoor effects—an advancement over prior DNN-based analyses that rely only on parameter-space linear mode connectivity.

3. Principled method (LFD).
The proposed defense—explicitly encouraging divergence in layer-wise feature representations—is conceptually clear, well-grounded in the LLFC analysis, and relatively easy to integrate with existing training pipelines.

4. Strong empirical evidence.
Comprehensive experiments across datasets and attacks demonstrate consistent improvements in robustness metrics, with ablation studies showing each loss component’s contribution.

5. Clear writing and reproducibility.
The manuscript is well-structured, follows ICLR conventions, and provides detailed implementation information with open-source code.

**Weaknesses:**

1. Limited novelty of defense formulation.
While the idea of maximizing feature divergence is reasonable, it may be viewed as an extension of existing regularization methods (e.g., feature orthogonalization, representation disentanglement). The theoretical justification of why this ensures durable robustness could be deepened.

2. Restricted attack scope.
Experiments only consider dirty-label backdoor attacks. The method’s effectiveness against clean-label attacks or more adaptive triggers is untested (acknowledged by the authors).

3. LLFC computation cost and scalability.
The paper does not discuss the computational overhead of LLFC-based loss during training, especially on large graphs (e.g., OGB datasets). A runtime or memory analysis would strengthen practical relevance.

4. Lack of theoretical analysis or guarantees.
The defense remains empirically motivated. There is no theoretical argument connecting feature divergence maximization to guaranteed reduction in feature-space backdoor correlation.

5. Missing visualization or qualitative results.
Some intuitive visualization of feature-space shifts (e.g., via t-SNE or cosine-similarity maps) could make the concept of “layer-wise divergence” more tangible.

**Questions:**

N/A

---

> ### Author Response · Authors · 2025-11-23
>
> Dear Reviewer **vCbK**,
>
> We sincerely appreciate your valuable and constructive feedback, which have helped us further improve the quality and clarity of the manuscript. We have carefully considered every comment, and below is our point-by-point response:
>
> ## W1 & W4: Further explanation of LFD
> Regarding the reviewer's questions about LFD, we provide further analysis and clarification here. LFD is not a traditional general regularization method. It is based on the theoretical concepts of LLFC, transforming LLFC from an analytical tool into a defensive target. LFD is a supervised, directed optimization training scheme that guides the defense model to accurately remove backdoor-related features by maximizing the difference between backdoor features and the model itself, while preserving benign classification features. Currently, the effectiveness of LFD is experiment-driven, but it is built on a solid theoretical foundation of LMC and LLFC. LMC theory posits that if two models are linearly connected in the parameter space, they tend to reside in the same basin and therefore share similar functional behaviors. Extending this to the backdoor and defense models, if the defense model easily exhibits backdoor behavior, it means that it and the original backdoor model are still located in a similar basin. Our theoretical intuition for LFD is that to truly remove the backdoor, the defensive model must be forced to move as far away as possible from the backdoor basin during the optimization process, and LLFC provides a measure of the degree of this distance. Furthermore, we have added some visualized experimental results in the Appendix of the revised paper. The t-SNE figure shows the final output feature space of the defense model. The results show that LFD pushes the poisoned nodes as far away from their original target labels as possible, creating a relatively clear isolation.

---

> ### Author Response · Authors · 2025-11-23
>
> ## W2: About clean-label setting
> Regarding the limitations mentioned by the reviewer under the clean-label setting, we further explain here. Firstly, our proposed poisoned node detection algorithm based on feature perturbation is currently only applicable to dirty-label. This is because its principle is to use the variance of the prediction results brought about by feature perturbation to distinguish between poisoned and clean nodes. However, under the clean-label setting, poisoned nodes and the target class share the same label, thus limiting the algorithm under this setting. But our method can be **generalized to the clean-label setting using the second-stage feature divergence training based on LLFC**. Specifically, the purpose of the first stage is to identify possible poisoned and clean nodes from the given data. As long as the corresponding sets of nodes of different categories are obtained, a more robust defense model can be trained through the divergence principle of the second stage. Therefore, the poisoned node detection algorithm of the first stage is actually replaceable. To verify the effectiveness of our statement, we selected the poisoned node sets identified by **RIGBD** and **DShield** and trained the defense on the LFD method. The experimental results are shown in the table below. It can be seen that the defense after using LFD is more robust than the defense of these methods themselves.
>
> | Method | Defense | Dataset | | | UGBA |  | | | GTA |  |
> | :--- | :--- | :---: | :---: | :---: | :---: | :---: | :---: | :---: | :---: | :---: |
> | | | | | CA | ASR | RA-ASR | | CA | ASR | RA-ASR |
> | **GCN** | RIGBD | **Cora** | | 81.85 | 0.40 | 40.24 | | 80.37 | 1.07 | 22.75 |
> | | RIGBD-LLFC (Dirty) | | | 83.81 | 0.30 | **31.08** | | 81.11 | 1.82 | **17.15** |
> | | Dshield | | | 82.22 | 0.40 | 86.45 | | 80.37 | 3.32 | 43.68 |
> | | Dshield-LLFC (Dirty) | | | 83.12 | 0.00 | **44.62** | | 80.37 | 3.11 | **33.07** |
> | | | | | | | | | | | |
> | | RIGBD | **PubMed** | | 83.51 | 2.14 | 22.57 | | 84.32 | 3.50 | 55.13 |
> | | RIGBD-LLFC (Dirty) | | | 83.74 | 3.27 | **10.21** | | 83.29 | 3.66 | **49.10** |
> | | Dshield | | | 83.87 | 2.97 | 86.41 | | 83.61 | 0.80 | **1.83** |
> | | Dshield-LLFC (Dirty) | | | 83.54 | 2.56 | **28.14** | | 83.54 | 0.80 | 1.84 |
> | | | | | | | | | | | |
>
> Furthermore, the **DShield** method can also be implemented with a clean-label setting. Therefore, we also conducted related experiments using the GCBA attack. As can be seen from the results in the table below, for clean-label backdoor attacks, LFD training also outperforms its original performance. In conclusion, although our method cannot be directly used with a clean-label setting, it can serve as a **general training paradigm**, directly applied after various poisoned node identification algorithms to further improve the robustness of the defense model.
>
> | Method | Defense | Dataset | | | GCBA |  |
> | :--- | :--- | :---: | :---: | :---: | :---: | :---: |
> | | | | | CA | ASR | RA-ASR |
> | **GCN** | Dshield | **Cora** | | 82.59 | 2.76 | 52.11 |
> | | Dshield-LLFC (Clean) | | | 81.89 | 2.81 | **15.54** |
> | | | | | | | |
> | | Dshield | **PubMed** | | 84.23 | 3.07 | 37.66 |
> | | Dshield-LLFC (Clean) | | | 83.96 | 3.04 | **19.63** |
> | | | | | | | | | | | |

---

> ### Author Response · Authors · 2025-11-23
>
> ## W3: About computation cost and scalability
> The reviewer considered the computational cost analysis essential, and we fully agree with the suggestion. We have added experiments, and the relevant results are shown in the table below. The table primarily displays the training and testing times for different defense methods, as well as the maximum memory usage during training. Although our LFD method outperforms the standard model and RIGBD in terms of training time and memory usage, these costs can be considered one-time expenses in the defense process. Once LFD is trained, its computational cost during the inference phase is close to that of the standard GCN model and lower than other defense methods.
>
> | Defense | Dataset | UGBA | | | GTA | | | DPGBA | |
> | :--- | :---: | :---: | :---: | :---: | :---: | :---: | :---: | :---: | :---: |
> | | | Train/Test_time | Memory | | Train/Test_time | Memory | | Train/Test_time | Memory |
> | **Clean Model** | **Cora** | 0.97s/0.65s | 0.0378GB | | 1.00s/0.64s | 0.0452GB | | 1.03s/0.65s | 0.0377GB |
> | **RIGBD** | | 4.00s/1.86s | 0.3295GB | | 5.96s/1.88s | 0.3427GB | | 3.53s/1.04s | 0.3581GB |
> | **DShield** | | 108.00s/2.80s | 1.3794GB | | 115.51s/3.81s | 1.6733GB | | 67.65s/2.76s | 1.5937GB |
> | **LFD(Ours)** | | 9.8s/0.79s | 0.3454GB | | 7.14s/0.62s | 0.3825GB | | 5.69s/0.79s | 0.4671GB |
> | | | | | | | | | | |
> | **Clean Model** | **PubMed** | 3.04s/1.86s | 0.1061GB | | 3.13s/1.84s | 0.1154GB | | 3.27s/1.80s | 0.1306GB |
> | **RIGBD** | | 14.47s/2.55s | 0.4616GB | | 16.35s/3.66s | 0.4822GB | | 17.46s/3.14s | 0.5186GB |
> | **DShield** | | 126.84s/5.04s | 1.5477GB | | 126.48s/5.75s | 1.6943GB | | 116.45s/5.50s | 1.6854GB |
> | **LFD(Ours)** | | 19.56s/2.06s | 0.4983GB | | 19.52s/2.97s | 0.5559GB | | 19.30s/2.22s | 0.5560GB |
> | | | | | | | | | | |
> | **Clean Model** | **OGB-arxiv**| 15.13s/51.46s | 1.0222GB | | 15.12s/64.36s | 1.0247GB | | 15.12s/61.43s | 1.0231GB |
> | **RIGBD** | | 21.13s/74.40s | 2.8764GB | | 21.37s/71.48s | 2.7842GB | | 20.23s/73.76s | 2.7628GB |
> | **DShield** | | 920.39s/85.19s | 4.9430GB | | 884.26s/96.16s | 4.9484GB | | 953.05s/117.26s | 4.9362GB |
> | **LFD(Ours)** | | 25.76s/61.62s | 2.9437GB | | 32.74s/67.22s | 2.9792GB | | 24.17s/63.74s | 2.8241GB |
> | | | | | | | | | | |
>
>
> We sincerely apologize for omitting the experimental results on the large-scale dataset OGB-arxiv in the appendix. The relevant results are shown in the table below. We will also add these results to Table 1 in the revised paper. The results demonstrate that on large datasets like OGB-arxiv, our proposed LFD achieves a low attack success rate while maintaining clean accuracy. Furthermore, RA-ASR outperforms other defense methods even after a redirecting attack, indicating that LFD plays a significant role in addressing the surface security problem of defense models.
>
> | Defense | Dataset |  | UGBA |  | | | GTA |  | | | DPGBA |  |
> | :--- | :---: | :---: | :---: | :---: | :---: | :---: | :---: | :---: | :---: | :---: | :---: | :---: |
> | | | CA | ASR | RA-ASR | | CA | ASR | RA-ASR | | CA | ASR | RA-ASR |
> | **Clean Model** | **OGB-arxiv** | 60.41 | 11.42 | 20.69 | | 60.38 | 0.84 | 13.58 | | 60.36 | 8.72 | 13.80 |
> | **No Defense** | | 59.32 | 99.79 | 79.55 | | 60.07 | 94.85 | 85.44 | | 59.77 | 93.60 | 84.53 |
> | **OP Model** | | 52.88 | 0.97 | 1.02 | | 58.44 | 0.00 | 2.43 | | 58.60 | 1.07 | 5.48 |
> | **RIGBD** | | 60.43 | 1.02 | 27.28 | | 60.66 | 7.23 | 17.19 | | 61.04 | 5.10 | 20.34 |
> | **DShield** | | 58.95 | 3.09 | 11.18 | | 57.71 | 6.58 | 24.04 | | 58.81 | 2.42 | 16.89 |
> | **LFD(Ours)** | | 59.94 | 0.40 | **4.80** | | 58.89 | 6.29 | **12.87** | | 59.22 | 3.61 | **9.86** |
> | | | | | | | | | | | | | |

---

> ### Author Response · Authors · 2025-11-23
>
> ## W5: Visualization results
> We thank the reviewer for the suggestions regarding the visualization analysis and have added the visualization results to the Appendix of the revised paper. We analyzed and compared the backdoor model, the OP model representing the theoretical upper bound of defense, RIGBD, DShield, and our LFD method. We visualized the t-SNE results of the node feature embeddings in the final output layer of the above models. The results show that the poisoned nodes (black) in the backdoor model are concentrated in the target class (orange) region, indicating that the backdoor attack is quite powerful. All four defense methods pushed the poisoned nodes away from the target class to some extent. Among them, OP, as a theoretical defense model, has complete knowledge of the backdoor, so it returns the poisoned nodes to the original cluster while pushing them away from the target class. The RIGBD defense method is generally scattered, and the decision boundary is unstable, where the residual backdoor features are easily reactivated. Although the DShield method also maps the poisoned nodes back to their original category, it mainly ignores the influence of the trigger through semantic and attribute differences, and does not fundamentally sever the mapping from the trigger to the target label. Therefore, some poisoned nodes are still very close to the target cluster. The LFD method utilizes the principle of hierarchical feature divergence to forcibly isolate poisoned nodes into a cluster that is as far away from the target label as possible, thus creating a greater security margin for the defense model.
>
>
> ## Conclusion
> We hope our response above has resolved your issue. If you have any further questions or details about the paper, we look forward to your suggestions. Some visualization results have been directly updated in Appendix of the revised paper; please feel free to view them if needed. We are currently working diligently to revise and improve the overall content and layout of the paper and will upload the final version as soon as possible.
>
> Best,
>
> Authors

---

> ### Author Response · Authors · 2025-11-27
> **Sincerely looking forward to your reply!**
>
> Dear Reviewer **vCbK**,
>
> We hope this message finds you well.
>
> We have carefully addressed your previous comments and made substantial revisions to our manuscript, as detailed in the responses and the revised PDF file (Text marked in blue).
>
> As the discussion period is approaching its deadline, we look forward to your feedback on the revised submission at your earliest convenience. Your insights and opinions are crucial for us to further improve the quality of the manuscript, and we greatly value the opportunity to continue communicating with you.
>
> Thanks again for your time and effort!
>
> Best,
>
> Authors

---

### Official Review · Reviewer_D7qM · 2025-10-31

**Soundness:** 2
**Presentation:** 2
**Contribution:** 2
**Rating:** 4
**Confidence:** 3

**Summary:**

This paper investigates the robustness of graph neural network (GNN) backdoor defenses and introduces a new defense strategy termed Layer-wise Feature Divergence (LFD). The authors first show that many existing backdoor defense methods for GNNs, despite appearing successful, remain fragile — as fine-tuning with even a small portion of poisoned data can reactivate hidden backdoor features. They attribute this fragility to residual feature similarity between the purified and the original backdoored models, analyzed via Layer-wise Linear Feature Connectivity (LLFC). To counteract this, the proposed LFD method enforces divergence between layer-wise features of the purified model and the backdoored one, aiming to achieve stronger robustness. The paper provides experimental results on standard datasets and compares against several existing defenses (RIGBD, DShield), reporting superior robustness under retraining attacks.

**Strengths:**

+ Provides the first empirical study on the robustness of graph backdoor defenses under reactivation attacks.
+ Offers clear empirical evidence that existing defenses can be easily re-poisoned, highlighting an underexplored vulnerability.

**Weaknesses:**

- Limited novelty — LFD is a direct adaptation of existing LMC ideas from DNNs, without deeper theoretical insights.
- Evaluation lacks breadth and rigor: only small datasets, missing modern or large-scale baselines, and no statistical testing.
- Heuristic method design — hyperparameter choices, loss weighting, and poisoned-node selection are ad-hoc and unexplained.
- Clean-label and black-box attack settings ignored, which weakens practical relevance.

**Questions:**

1. Is the divergence enforced equally across all layers, or do certain layers (e.g., deeper GCN layers) contribute more to robustness? Would focusing the divergence on high-level representations yield better results?

2. The paper uses LLFC mainly as a visualization tool — is there any quantitative evidence that LLFC distance actually correlates with robustness metrics such as RA-ASR?

3. Can the proposed method generalize beyond dirty-label settings? Specifically, how would it perform under clean-label or federated graph backdoor attacks where trigger visibility and label control are limited?

4. How would LFD handle structural perturbations — for instance, attacks that modify edges rather than node features? Would the same feature divergence principle apply?

5. What is the computational cost of computing layer-wise divergence, especially for deeper GNNs or large-scale datasets?

6. Finally, the core assumption that maximizing feature divergence leads to true backdoor removal seems intuitive but unproven — could it instead just push the model into a different yet still vulnerable region of the feature space?

---

> ### Author Response · Authors · 2025-11-23
>
> Dear Reviewer **D7qM**,
>
> We sincerely appreciate your valuable and constructive feedback, which have helped us further improve the quality and clarity of the manuscript. We have carefully considered every comment, and below is our point-by-point response:
>
> ## W1 & Q6：Further explanation of LFD
> Regarding the reviewer's questions about LFD, we provide further analysis and clarification here. While the concept of LLFC does indeed originate from the analysis of LMC in the DNN field, our work focuses on transforming LLFC from an analytical tool into a defensive target. Furthermore, considering that the message passing mechanism unique to GNNs causes backdoor features to diffuse with inter-layer aggregation, we propose a layer-by-layer constraint based on feature divergence. Related work in DNNs mainly utilizes LMC starting from the output of the last layer to guide model optimization. Our approach, however, combines the unique structure of GNNs, breaking feature connectivity at each layer to block the aggregation of backdoor features.
>
> To address the reviewer's concern that the model might be pushed into still vulnerable regions, our proposed retuning attack (RA) method is essentially a detection technique that searches for residual backdoors within the neighborhood of the model parameters. If **LFD** is merely pushed into a still vulnerable region, its backdoors will still be easily activated after RA, resulting in a large RA-ASR. However, experimental data shows that the RA-ASR of LFD is smaller, indicating that the feature region reached by LFD is not only far from the original backdoor model but also robust near local minima, with no easily activated residual backdoors. To further quantify the relationship between LLFC and RA-ASR, we present a quantitative relationship between the two in **Q2**, further demonstrating that higher feature divergence corresponds to fewer residual backdoors.

---

> ### Author Response · Authors · 2025-11-23
>
> ## W2 & Q5: Experiments on large-scale dataset
> We sincerely apologize for omitting the experimental results on the large-scale dataset OGB-arxiv in the appendix. The relevant results are shown in the table below. We will also add these results to Table 1 in the revised paper. The results demonstrate that on large datasets like OGB-arxiv, our proposed LFD achieves a low attack success rate while maintaining clean accuracy. Furthermore, RA-ASR outperforms other defense methods even after a redirecting attack, indicating that LFD plays a significant role in addressing the surface security problem of defense models. Regarding the reviewer's comment about the lack of statistical tests in the experiments, we sincerely apologize for not explicitly addressing this in the experimental section. For all results in the paper, we ran all experiments three times and averaged all results.
>
> | Defense | Dataset | | UGBA |  | | | GTA |  | | | DPGBA |  |
> | :--- | :---: | :---: | :---: | :---: | :---: | :---: | :---: | :---: | :---: | :---: | :---: | :---: |
> | | | CA | ASR | RA-ASR | | CA | ASR | RA-ASR | | CA | ASR | RA-ASR |
> | **Clean Model** | **OGB-arxiv** | 60.41 | 11.42 | 20.69 | | 60.38 | 0.84 | 13.58 | | 60.36 | 8.72 | 13.80 |
> | **No Defense** | | 59.32 | 99.79 | 79.55 | | 60.07 | 94.85 | 85.44 | | 59.77 | 93.60 | 84.53 |
> | **OP Model** | | 52.88 | 0.97 | 1.02 | | 58.44 | 0.00 | 2.43 | | 58.60 | 1.07 | 5.48 |
> | **RIGBD** | | 60.43 | 1.02 | 27.28 | | 60.66 | 7.23 | 17.19 | | 61.04 | 5.10 | 20.34 |
> | **DShield** | | 58.95 | 3.09 | 11.18 | | 57.71 | 6.58 | 24.04 | | 58.81 | 2.42 | 16.89 |
> | **LFD(Ours)** | | 59.94 | 0.40 | **4.80** | | 58.89 | 6.29 | **12.87** | | 59.22 | 3.61 | **9.86** |
> | | | | | | | | | | | | | |
>
> The reviewer considered the computational cost analysis essential, and we fully agree with the suggestion. We have added experiments, and the relevant results are shown in the table below. The table primarily displays the training and testing times for different defense methods, as well as the maximum memory usage during training. Although our LFD method outperforms the standard model and RIGBD in terms of training time and memory usage, these costs can be considered one-time expenses in the defense process. Once LFD is trained, its computational cost during the inference phase is close to that of the standard GCN model and lower than other defense methods.
>
> | Defense | Dataset | UGBA | | | GTA | | | DPGBA | |
> | :--- | :---: | :---: | :---: | :---: | :---: | :---: | :---: | :---: | :---: |
> | | | Train/Test_time | Memory | | Train/Test_time | Memory | | Train/Test_time | Memory |
> | **Clean Model** | **Cora** | 0.97s/0.65s | 0.0378GB | | 1.00s/0.64s | 0.0452GB | | 1.03s/0.65s | 0.0377GB |
> | **RIGBD** | | 4.00s/1.86s | 0.3295GB | | 5.96s/1.88s | 0.3427GB | | 3.53s/1.04s | 0.3581GB |
> | **DShield** | | 108.00s/2.80s | 1.3794GB | | 115.51s/3.81s | 1.6733GB | | 67.65s/2.76s | 1.5937GB |
> | **LFD(Ours)** | | 9.8s/0.79s | 0.3454GB | | 7.14s/0.62s | 0.3825GB | | 5.69s/0.79s | 0.4671GB |
> | | | | | | | | | | |
> | **Clean Model** | **PubMed** | 3.04s/1.86s | 0.1061GB | | 3.13s/1.84s | 0.1154GB | | 3.27s/1.80s | 0.1306GB |
> | **RIGBD** | | 14.47s/2.55s | 0.4616GB | | 16.35s/3.66s | 0.4822GB | | 17.46s/3.14s | 0.5186GB |
> | **DShield** | | 126.84s/5.04s | 1.5477GB | | 126.48s/5.75s | 1.6943GB | | 116.45s/5.50s | 1.6854GB |
> | **LFD(Ours)** | | 19.56s/2.06s | 0.4983GB | | 19.52s/2.97s | 0.5559GB | | 19.30s/2.22s | 0.5560GB |
> | | | | | | | | | | |
> | **Clean Model** | **OGB-arxiv**| 15.13s/51.46s | 1.0222GB | | 15.12s/64.36s | 1.0247GB | | 15.12s/61.43s | 1.0231GB |
> | **RIGBD** | | 21.13s/74.40s | 2.8764GB | | 21.37s/71.48s | 2.7842GB | | 20.23s/73.76s | 2.7628GB |
> | **DShield** | | 920.39s/85.19s | 4.9430GB | | 884.26s/96.16s | 4.9484GB | | 953.05s/117.26s | 4.9362GB |
> | **LFD(Ours)** | | 25.76s/61.62s | 2.9437GB | | 32.74s/67.22s | 2.9792GB | | 24.17s/63.74s | 2.8241GB |
> | | | | | | | | | | |

---

> ### Author Response · Authors · 2025-11-23
>
> ## W3: Additional explanation on heuristic methods
> We thank the reviewer for the suggestion. We have conducted a detailed ablation study on the weights of the loss terms. The weights of the three loss terms in the loss function are denoted as **loss_clean**, **loss_suppress**, and **loss_llfc**, respectively. We fixed one parameter and varied the other across $[0.1, 1.0, 10.0]$ to observe the impact on CA, ASR and RA-ASR. We provided some experimental results, as shown in the table below, and added the visualization results to the Appendix of the revised paper.
>
> | Dataset | Attack | loss_clean | loss_suppress | loss_llfc | | CA | ASR | RA-ASR |
> | :--- | :--- | :---: | :---: | :---: | :---: | :---: | :---: | :---: |
> | **Cora** | **UGBA** | 1.0 | 1.0 | 0.1 | | 83.33 | 4.79 | 74.50 |
> | | | 1.0 | 1.0 | 1.0 | | 82.96 | 6.51 | 63.35 |
> | | | 1.0 | 1.0 | 10.0 | | 83.70 | 10.62 | 43.03 |
> | | | 1.0 | 0.1 | 1.0 | | 83.70 | 18.00 | 58.17 |
> | | | 1.0 | 1.0 | 1.0 | | 81.48 | 16.94 | 64.54 |
> | | | 1.0 | 10.0 | 1.0 | | 82.59 | 15.77 | 70.29 |
> | | | | | | | | | |
> | **OGB-arxiv** | **GTA** | 1.0 | 1.0 | 0.1 | | 61.21 | 18.07 | 26.69 |
> | | | 1.0 | 1.0| 1.0 | | 61.10 | 18.64 | 25.98 |
> | | | 1.0| 1.0 | 10.0 | | 60.62 | 20.21 | 21.55 |
> | | | 1.0 | 0.1 | 1.0 | | 61.07 | 19.64 | 16.25 |
> | | | 1.0 | 1.0 | 1.0 | | 60.56 | 19.47 | 16.56 |
> | | | 1.0 | 10.0 | 1.0 | | 59.91 | 18.38 | 17.36 |
>
> **Robustness of Clean Accuracy (CA):** Under all parameter settings, CA is generally stable and performs similarly to the clean model. This indicates that our defenses do not affect the model's utility on benign tasks.
>
> **Role of Loss_llfc (Feature Divergence):** As this value increases, RA-ASR can be significantly reduced. This confirms that the LLFC term is the main driver of robust defense, effectively establishing a large feature distance with the backdoor model, thereby preventing the reactivation of backdoor features.
>
> **Role of Loss_suppress (Attack Suppression):** Increasing this value can effectively reduce ASR, but has no positive effect on RA-ASR. This suggests that this term may primarily focus on reducing the success rate of attacks by improving output probability, rather than defending against attacks from the perspective of distance from internal features.
>
> These findings reveal a trade-off: loss_suppress ensures low ASR, while loss_llfc ensures low RA-ASR. Based on this analysis, we selected the configuration of loss_suppress=2.0 and loss_suppress=5.0 for our main experiments to achieve an optimal balance.

---

> ### Author Response · Authors · 2025-11-23
>
> Regarding the reviewer‘s question about the poisoned node detection algorithm, we provide further explanation and supplementation here. Our algorithm primarily utilizes the difference in the impact of random perturbations of neighbor features on the prediction results to distinguish between poisoned and clean nodes. For clean nodes, the prediction result is highly dependent on the feature aggregation of benign neighbors; therefore, changes in the features of surrounding neighbors will significantly affect the prediction result. For poisoned nodes, the prediction result is highly dependent on triggers, and the mapping from triggers to target labels is generally overfitted. Therefore, even if a certain perturbation is added to the neighbor triggers, the change in the prediction result is not significant. To intuitively illustrate the effectiveness of the poisoned node detection algorithm, we supplemented the following experiments. The overall defense mode of the graph backdoor defense method **RIGBD** is the same as our **LFD**, mainly identifying poisoned nodes in the first stage. Therefore, we selected three metrics **recall**, **precision**, and **F1-score** to compare the recognition performance of the two methods. As can be seen from the results in the table below, although our detection algorithm is slightly weaker than RIGBD, in the second stage of defense training, LFD's defense robustness is far superior to RIGBD. This further demonstrates the key role of feature divergence optimization training.
>
> | Datasets | Attacks | | | | RIGBD | |  | | | | LFD | |  |
> | :--- | :--- | :---: | :---: | :---: | :---: | :---: | :---: | :---: | :---: | :---: | :---: | :---: | :---: |
> | | | | Recall | Precision | F1-score | ASR | RA-ASR | | Recall | Precision | F1-score | ASR | RA-ASR |
> | **Cora** | UGBA | | 0.968 | 1.000 | 0.984 | 0.40 | 40.24 | | 0.900 | 1.000 | 0.947 | 0.00 | **15.54** |
> | | GTA | | 0.912 | 0.940 | 0.926 | 1.07 | **22.75** | | 0.875 | 1.000 | 0.933 | 6.10 | 31.07 |
> | | DPGBA | | 1.000 | 1.000 | 1.000 | 6.64 | 23.94 | | 0.869 | 0.914 | 0.891 | 9.16 | **21.91** |
> | | | | | | | | | | | | | | |
> | **OGB-arxiv**| UGBA | | 0.956 | 0.963 | 0.959 | 1.02 | 27.28 | | 0.852 | 0.910 | 0.880 | 0.40 | **4.80** |
> | | GTA | | 0.849 | 0.903 | 0.875 | 7.23 | 17.19 | | 0.811 | 0.858 | 0.834 | 6.29 | **12.87** |
> | | DPGBA | | 0.905 | 0.939 | 0.922 | 5.10 | 20.34 | | 0.862 | 0.851 | 0.856 | 3.61 | **9.86** |
> | | | | | | | | | | | | | | |
> | **PubMed** | UGBA | | 0.887 | 0.966 | 0.925 | 2.14 | 22.57 | | 0.812 | 0.957 | 0.879 | 0.67 | **11.11** |
> | | GTA | | 0.820 | 0.905 | 0.860 | 3.50 | 55.13 | | 1.000 | 0.976 | 0.988 | 0.80 | **1.90** |
> | | DPGBA | | 0.850 | 0.910 | 0.879 | 3.39 | 8.33 | | 0.785 | 0.850 | 0.816 | 8.78 | **7.65** |
> | | | | | | | | | | | | | | |

---

> ### Author Response · Authors · 2025-11-23
>
> ## W4 & Q3: About clean-label and black-box setting
> Regarding the limitations mentioned by the reviewer under the clean-label setting, we further explain here. Firstly, our proposed poisoned node detection algorithm based on feature perturbation is currently only applicable to dirty-label. This is because its principle is to use the variance of the prediction results brought about by feature perturbation to distinguish between poisoned and clean nodes. However, under the clean-label setting, poisoned nodes and the target class share the same label, thus limiting the algorithm under this setting. But our method can be **generalized to the clean-label setting using the second-stage feature divergence training based on LLFC**. Specifically, the purpose of the first stage is to identify possible poisoned and clean nodes from the given data. As long as the corresponding sets of nodes of different categories are obtained, a more robust defense model can be trained through the divergence principle of the second stage. Therefore, the poisoned node detection algorithm of the first stage is actually replaceable. To verify the effectiveness of our statement, we selected the poisoned node sets identified by **RIGBD** and **DShield** and trained the defense on the LFD method. The experimental results are shown in the table below. It can be seen that the defense after using LFD is more robust than the defense of these methods themselves.
>
> | | Defense | Dataset | | | UGBA |  | | | GTA |  |
> | :--- | :--- | :---: | :---: | :---: | :---: | :---: | :---: | :---: | :---: | :---: |
> | | | | | CA | ASR | RA-ASR | | CA | ASR | RA-ASR |
> | | RIGBD | **Cora** | | 81.85 | 0.40 | 40.24 | | 80.37 | 1.07 | 22.75 |
> | | RIGBD-LLFC (Dirty) | | | 83.81 | 0.30 | **31.08** | | 81.11 | 1.82 | **17.15** |
> | | Dshield | | | 82.22 | 0.40 | 86.45 | | 80.37 | 3.32 | 43.68 |
> | | Dshield-LLFC (Dirty) | | | 83.12 | 0.00 | **44.62** | | 80.37 | 3.11 | **33.07** |
> | | | | | | | | | | | |
> | | RIGBD | **PubMed** | | 83.51 | 2.14 | 22.57 | | 84.32 | 3.50 | 55.13 |
> | | RIGBD-LLFC (Dirty) | | | 83.74 | 3.27 | **10.21** | | 83.29 | 3.66 | **49.10** |
> | | Dshield | | | 83.87 | 2.97 | 86.41 | | 83.61 | 0.80 | **1.83** |
> | | Dshield-LLFC (Dirty) | | | 83.54 | 2.56 | **28.14** | | 83.54 | 0.80 | 1.84 |
> | | | | | | | | | | | |
>
> Furthermore, the **DShield** method can also be implemented with a clean-label setting. Therefore, we also conducted related experiments using the GCBA attack. As can be seen from the results in the table below, for clean-label backdoor attacks, LFD training also outperforms its original performance. In conclusion, although our method cannot be directly used with a clean-label setting, it can serve as a **general training paradigm**, directly applied after various poisoned node identification algorithms (including dirty-label, clean-label, federated graph and black-box settings), to further improve the robustness of the defense model.
>
> |  | Defense | Dataset | | | GCBA |  |
> | :--- | :--- | :---: | :---: | :---: | :---: | :---: |
> |  | | | | CA | ASR | RA-ASR |
> |  | Dshield | **Cora** | | 82.59 | 2.76 | 52.11 |
> | | Dshield-LLFC (Clean) | | | 81.89 | 2.81 | **15.54** |
> | | | | | | | |
> | | Dshield | **PubMed** | | 84.23 | 3.07 | 37.66 |
> | | Dshield-LLFC (Clean) | | | 83.96 | 3.04 | **19.63** |
> | | | | | | | | | | | |

---

> ### Author Response · Authors · 2025-11-23
>
> ## Q1: Analysis of divergence between layers
> Thanks for the meaningful question raised by the reviewer. Our method is to take the average of the output feature divergence of the two layers of GCN as an optimization term of the model. According to the reviewer's suggestions, we added some ablation experiments. Specifically, we selected only the output feature of a certain layer as an optimization term, and analyzed the actual effect of applying divergence optimization to different layers according to the three indicators CA, ASR and RA-ASR. The results are shown in the table below. It can be seen that the overall effect of using the average of the two layers of features is better. In addition, we also found that the feature divergence in the first layer is better than that in the second layer. This is because for models with very few layers like GCN, the first layer is usually used for original feature extraction and is far away from backdoor features. Therefore, applying divergence optimization here can effectively keep the defense model away from the backdoor. The output of the second layer is generally used to achieve the task objective, so the effect of divergence is generally average. Regarding the reviewers' suggestion that "the effect may be better in deeper layers", the paper [1] reflects this. For models with more layers, the feature divergence of the middle layer is larger, while the divergence of the last layer is usually very small. However, conventional GCN models typically use only a two-layer structure, with the second layer serving as the last layer, primarily for achieving the task objectives.
>
> | Dataset | Layer1 | Layer2 | | | UGBA |  | | | GTA |  |
> | :--- | :---: | :---: | :---: | :---: | :---: | :---: | :---: | :---: | :---: | :---: |
> | | | | | CA | ASR | RA-ASR | | CA | ASR | RA-ASR |
> | **Cora** | ✓ | | | 81.11 | 7.99 | 20.32 | | 82.22 | 20.26 | 57.71 |
> | | | ✓ | | 82.12 | 12.51 | 27.49 | | 81.11 | 31.65 | 74.90 |
> | | ✓ | ✓ | | **83.70** | **0.00** | **15.54** | | **83.70** | **6.10** | **31.07** |
> | | | | | | | | | | | |
> | **PubMed** | ✓ | | | 83.41 | 5.74 | 17.05 | | 85.74 | 2.67 | 16.54 |
> | | | ✓ | | 83.26 | 4.19 | 18.64 | | 85.34 | 3.11 | 29.76 |
> | | ✓ | ✓ | | **84.88** | **0.67** | **11.11** | | **85.03** | **0.80** | **1.90** |
> | | | | | | | | | | | |
> | **OGB-arxiv**| ✓ | | | 60.23 | 2.97 | 6.64 | | 60.25 | 19.75 | 27.67 |
> | | | ✓ | | **60.93** | 3.11 | 6.73 | | 60.69 | 20.69 | 27.45 |
> | | ✓ | ✓ | | 59.94 | **0.40** | **4.80** | | **58.89** | **6.29** | **12.87** |
> | | | | | | | | | | | |
>
> [1] Zhou, Zhanpeng, et al. "Going beyond linear mode connectivity: The layerwise linear feature connectivity." (NeurIPS 2023).

---

> ### Author Response · Authors · 2025-11-23
>
> ## Q2: Quantitative analysis between LLFC and RA-ASR
> We strongly agree with the reviewers' insightful suggestion to quantitatively analyze the LLFC distance and RA-ASR, which would more intuitively demonstrate the crucial role of increasing the hierarchical feature distance in improving the robustness of the defense model. We present the relevant experimental results in tabular form in this response for reviewers to easily review. The results are as follows:
>
> | Dataset | Defense | | UGBA |  | | | GTA |  | | | DPGBA |  |
> | :--- | :--- | :---: | :---: | :---: | :---: | :---: | :---: | :---: | :---: | :---: | :---: | :---: |
> | | | L1 | L2 | RA-ASR | | L1 | L2 | RA-ASR | | L1 | L2 | RA-ASR |
> | **Cora** | OP | 0.7332 | 0.1714 | 10.36 | | 0.6741 | 0.1584 | 13.55 | | 0.5189 | 0.1748 | 7.17 |
> | | RIGBD | 0.2294 | 0.1953 | 40.24 | | 0.4972 | 0.1746 | 22.75 | | 0.3396 | 0.1077 | 23.94 |
> | | Dshield | 0.3821 | 0.1710 | 86.45 | | 0.4021 | 0.1236 | 43.68 | | 0.2856 | 0.1219 | 25.10 |
> | | **LFD(Ours)** | 0.4164 | 0.1634 | 15.54 | | 0.5404 | 0.1526 | 31.07 | | 0.4291 | 0.1442 | 21.91 |
> | | | | | | | | | | | | | |
> | **PubMed** | OP | 0.9685 | 0.1827 | 7.72 | | 0.9932 | 0.1864 | 1.59 | | 0.7516 | 0.1950 | 0.80 |
> | | RIGBD | 0.6344 | 0.2496 | 22.57 | | 0.3313 | 0.0374 | 55.13 | | 0.5298 | 0.0982 | 8.33 |
> | | Dshield | 0.5853 | 0.2415 | 86.41 | | 0.7371 | 0.2092 | 1.83 | | 0.5392 | 0.1047 | 8.07 |
> | | **LFD(Ours)** | 0.7348 | 0.2755 | 11.11 | | 0.2914 | 0.3130 | 1.90 | | 0.5709 | 0.1176 | 7.65 |
> | | | | | | | | | | | | | |
> | **OGB-arxiv** | OP | 0.6920 | 0.1368 | 1.02 | | 0.6719 | 0.1545 | 2.43 | | 0.5345 | 0.1274 | 5.48 |
> | | RIGBD | 0.3513 | 0.0773 | 27.28 | | 0.5338 | 0.0689 | 17.19 | | 0.2455 | 0.0599 | 20.34 |
> | | Dshield | 0.4184 | 0.0949 | 11.18 | | 0.4626 | 0.0186 | 24.04 | | 0.2812 | 0.0478 | 16.89 |
> | | **LFD(Ours)** | 0.5709 | 0.0961 | 4.80 | | 0.5203 | 0.0980 | 12.87 | | 0.3126 | 0.0651 | 9.86 |
> | | | | | | | | | | | | | |
>
> **Results Analysis:** L1 and L2 represent the LLFC distance between the two layers of the GCN, respectively, and RA-ASR represents the ASR of the defense model after being attacked by RA. It can be seen that under any dataset and attack method, the LLFC distance between the two layers is roughly inversely proportional to RA-ASR. That is, the greater the difference in inter-layer feature similarity between the defense model and the backdoor model, the smaller the increase in ASR after a retuning attack. This further corroborates the crucial role of increasing the layer feature distance in improving the robustness of the defense model. Furthermore, the results show that the value of L1 is generally greater than L2. This is because for a two-layer GCN defense model, the main function of the first layer is to extract data features while keeping those features as far away from the backdoor as possible. While the feature distance of the second layer decreases overall, this is because regardless of the defense method, their ultimate goal is the same: to maintain clean accuracy and reduce attack success rate; backdoor attack methods also need to maintain clean accuracy. Therefore, the LLFC distance of the second layer decreases overall.
>
> **Visualization result:** In addition to data analysis, we have added more intuitive visualizations to the revised paper. As shown in Appendix, the figure illustrates the approximate relationship between LLFC and RA-ASR for different defense methods. Taking any sub-figure as an example, the dots to the right of the horizontal line represent the LLFC distance of L1, and the crosses to the left represent the LLFC distance of L2. The horizontal axis represents the LLFC distance, and the vertical axis represents RA-ASR. It can be seen that OP, as the theoretical upper bound of the defense model, has the largest overall LLFC. At the same time, it has the smallest RA-ASR. Compared with the other two defense methods, our LFD method has a larger LLFC and a smaller RA-ASR. Overall, the approximate inverse relationship between LLFC and RA-ASR can also be seen.

---

> ### Author Response · Authors · 2025-11-23
>
> ## Q4: LFD for structural perturbations
> The feature divergence principle of LFD also applies to structural attacks. This is because in GNNs, the topology (edges) is not independent; it directly determines the feature representation of nodes through an aggregation mechanism. Any structural perturbation will ultimately map to changes in the node's feature space. Whether the attacker modifies features or edges, as long as these modifications attempt to mislead classification in the feature space through specific triggers, the feature divergence constraint of LFD will break this specific feature mapping, thus achieving defense. Therefore, the subsequent feature divergence optimization defense process still applies.
>
> ## Conclusion
> We hope our response above has resolved your issue. If you have any further questions or details about the paper, we look forward to your suggestions. Some visualization results have been directly updated in Appendix of the revised paper; please feel free to view them if needed. We are currently working diligently to revise and improve the overall content and layout of the paper and will upload the final version as soon as possible.
>
> Best,
>
> Authors

---

> ### Author Response · Authors · 2025-11-27
> **Sincerely looking forward to your reply!**
>
> Dear Reviewer **D7qM**,
>
> We hope this message finds you well.
>
> We have carefully addressed your previous comments and made substantial revisions to our manuscript, as detailed in the responses and the revised PDF file (Text marked in blue).
>
> As the discussion period is approaching its deadline, we look forward to your feedback on the revised submission at your earliest convenience. Your insights and opinions are crucial for us to further improve the quality of the manuscript, and we greatly value the opportunity to continue communicating with you.
>
> Thanks again for your time and effort!
>
> Best,
>
> Authors

---

### Official Review · Reviewer_LrTi · 2025-10-31

**Soundness:** 2
**Presentation:** 3
**Contribution:** 3
**Rating:** 4
**Confidence:** 4

**Summary:**

This paper presents the systematic robustness analysis of graph backdoor defenses, revealing that existing methods only achieve superficial security and can be easily reactivated. The authors propose a novel defense strategy called Layer-wise Feature Divergence (LFD), which enhances robustness by maximizing feature-space divergence from the backdoored model during retraining.

**Strengths:**

1)	First to systematically investigate the fragility of graph backdoor defenses.
2)	Leverages LLFC to explain vulnerabilities in the feature space.
3)	The paper is well-structured, clearly written, and supported by informative figures and tables.

**Weaknesses:**

1)	The core of LFD requires a clean, backdoored reference model to maximize feature divergence. This creates a fundamental paradox: in a real-world defense scenario, such a pristine backdoored model is unavailable. If the defender already possesses it, why not use it directly for training?

2)	The analysis of LLFC remains superficial, with limited theoretical contribution.

3)	The vulnerabilities of the poisoning node detection module have not been adequately discussed.

**Questions:**

1、It is unclear whether the method proposed by the author is applicable to large-scale graph data, such as the OGB-arxiv dataset mentioned in Section 6, as it was not demonstrated in the experiments.

2、From Table 2, it can be seen that in the clean node classification task, using only clean_loss actually performs worse compared to using w/o suppress_loss. The authors don't seem to have paid attention to this point. Does keeping the defense model away from the original backdoor model have a greater impact on the clean node classification task?

3、Weakness1.

4、Although the author introduced the LLFC tool, its usage is basically descriptive rather than diagnostic, without exploring which layers' features are most critical for backdoor recovery?

5、The authors acknowledge that LFD is only evaluated against dirty-label attacks. However, more advanced and stealthy clean-label attacks pose a significant threat【1】. Does the core assumption of LFD—that poisoned nodes rely on triggers and are insensitive to neighbors—hold under clean-label attacks? Would the defense mechanism of LFD be rendered ineffective in such scenarios?

---

> ### Author Response · Authors · 2025-11-23
>
> Dear Reviewer **LrTi**,
>
> We sincerely appreciate your valuable and constructive feedback, which have helped us further improve the quality and clarity of the manuscript. We have carefully considered every comment, and below is our point-by-point response:
>
> ## W1 & Q3: About backdoor model setting
> Thanks to the reviewer for pointing out this potential expression problem. Our backdoor model is a threat model in a standard **model purification** scenario. Under this setting, the knowledge possessed by the defender usually includes: white-box access to the threat model and some poisoned data (but the defender does not know the specific poisoning information). What the defender does is to train or fine-tune the threat model to disable its original backdoor, while ensuring that the performance on clean data is not affected. A series of existing model purification works are also based on this standard setting, such as [1] on DNN, and [2] and [3] on GNN.
>
> [1] Min, Rui, et al. "Uncovering, explaining, and mitigating the superficial safety of backdoor defense." (NeurIPS 2024).
>
> [2] Zhang, Zhiwei, et al. "Robustness-inspired defense against backdoor attacks on graph neural networks." (ICLR 2025).
>
> [3] Yu, Hao, et al. "Dshield: Defending against backdoor attacks on graph neural networks via discrepancy learning." (NDSS 2025).
>
>
> ## W2 & Q4: The analysis of LLFC
> Regarding the theoretical analysis of LLFC, we provide supplementary explanations here. Previous defense efforts primarily focused on reducing ASR (Attack Success Rate), successfully masking the backdoor in the final output layer. However, after a retuning attack, the backdoor is easily reactivated. Existing work fails to explain why the backdoor is prone to recurrence. We utilize LLFC to reveal for the first time that while existing SOTA defenses perform well in ASR, they remain highly connected to the backdoor model in intermediate layers. This connectivity manifests as high inter-layer feature similarity. Therefore, we transform LLFC from an analytical tool into a defensive objective, starting with the unique inter-layer message passing mechanism of GNNs, and training a more robust defense model with the optimization objective of maximizing feature divergence. In the Appendix of the revised paper, we have added visualizations of the quantitative relationship between LLFC distance and RA-ASR. The results also intuitively show that LLFC distance and RA-ASR are roughly inversely proportional. That is, the greater the difference in inter-layer feature similarity between the defense model and the backdoor model, the smaller the ASR increase after a redirecting attack. This further corroborates the crucial role of increasing the hierarchical feature distance in improving the robustness of the defense model.
>
> Based on your suggestions, we conducted further ablation experiments to explore the importance of feature divergence in different layers. Specifically, we selected only the output feature of a certain layer as an optimization term, and analyzed the actual effect of applying divergence optimization to different layers according to the three indicators CA, ASR and RA-ASR. The results are shown in the table below. It can be seen that the overall effect of using the average of the two layers of features is better. In addition, we also found that the feature divergence in the first layer is better than that in the second layer. This is because for models with very few layers like GCN, the first layer is usually used for original feature extraction and is far away from backdoor features. Therefore, applying divergence optimization here can effectively keep the defense model away from the backdoor. The output of the second layer is generally used to achieve the task objective, so the effect of divergence is generally average.
>
> | Dataset | Layer1 | Layer2 | | | UGBA |  | | | GTA |  |
> | :--- | :---: | :---: | :---: | :---: | :---: | :---: | :---: | :---: | :---: | :---: |
> | | | | | CA | ASR | RA-ASR | | CA | ASR | RA-ASR |
> | **Cora** | ✓ | | | 81.11 | 7.99 | 20.32 | | 82.22 | 20.26 | 57.71 |
> | | | ✓ | | 82.12 | 12.51 | 27.49 | | 81.11 | 31.65 | 74.90 |
> | | ✓ | ✓ | | **83.70** | **0.00** | **15.54** | | **83.70** | **6.10** | **31.07** |
> | | | | | | | | | | | |
> | **PubMed** | ✓ | | | 83.41 | 5.74 | 17.05 | | 85.74 | 2.67 | 16.54 |
> | | | ✓ | | 83.26 | 4.19 | 18.64 | | 85.34 | 3.11 | 29.76 |
> | | ✓ | ✓ | | **84.88** | **0.67** | **11.11** | | **85.03** | **0.80** | **1.90** |
> | | | | | | | | | | | |
> | **OGB-arxiv**| ✓ | | | 60.23 | 2.97 | 6.64 | | 60.25 | 19.75 | 27.67 |
> | | | ✓ | | **60.93** | 3.11 | 6.73 | | 60.69 | 20.69 | 27.45 |
> | | ✓ | ✓ | | 59.94 | **0.40** | **4.80** | | **58.89** | **6.29** | **12.87** |
> | | | | | | | | | | | |

---

> ### Author Response · Authors · 2025-11-23
>
> ## W3: About poisoned node detection algorithm
> Regarding the reviewer‘s question about the poisoned node detection algorithm, we provide further explanation and supplementation here. Our algorithm primarily utilizes the difference in the impact of random perturbations of neighbor features on the prediction results to distinguish between poisoned and clean nodes. For clean nodes, the prediction result is highly dependent on the feature aggregation of benign neighbors; therefore, changes in the features of surrounding neighbors will significantly affect the prediction result. For poisoned nodes, the prediction result is highly dependent on triggers, and the mapping from triggers to target labels is generally overfitted. Therefore, even if a certain perturbation is added to the neighbor triggers, the change in the prediction result is not significant. To intuitively illustrate the effectiveness of the poisoned node detection algorithm, we supplemented the following experiments. The overall defense mode of the graph backdoor defense method **RIGBD** is the same as our **LFD**, mainly identifying poisoned nodes in the first stage. Therefore, we selected three metrics **recall**, **precision**, and **F1-score** to compare the recognition performance of the two methods. As can be seen from the results in the table below, although our detection algorithm is slightly weaker than RIGBD, in the second stage of defense training, LFD's defense robustness is far superior to RIGBD. This further demonstrates the key role of feature divergence optimization training.
>
> | Datasets | Attacks | | | | RIGBD | |  | | | | LFD | |  |
> | :--- | :--- | :---: | :---: | :---: | :---: | :---: | :---: | :---: | :---: | :---: | :---: | :---: | :---: |
> | | | | Recall | Precision | F1-score | ASR | RA-ASR | | Recall | Precision | F1-score | ASR | RA-ASR |
> | **Cora** | UGBA | | 0.968 | 1.000 | 0.984 | 0.40 | 40.24 | | 0.900 | 1.000 | 0.947 | 0.00 | **15.54** |
> | | GTA | | 0.912 | 0.940 | 0.926 | 1.07 | **22.75** | | 0.875 | 1.000 | 0.933 | 6.10 | 31.07 |
> | | DPGBA | | 1.000 | 1.000 | 1.000 | 6.64 | 23.94 | | 0.869 | 0.914 | 0.891 | 9.16 | **21.91** |
> | | | | | | | | | | | | | | |
> | **OGB-arxiv**| UGBA | | 0.956 | 0.963 | 0.959 | 1.02 | 27.28 | | 0.852 | 0.910 | 0.880 | 0.40 | **4.80** |
> | | GTA | | 0.849 | 0.903 | 0.875 | 7.23 | 17.19 | | 0.811 | 0.858 | 0.834 | 6.29 | **12.87** |
> | | DPGBA | | 0.905 | 0.939 | 0.922 | 5.10 | 20.34 | | 0.862 | 0.851 | 0.856 | 3.61 | **9.86** |
> | | | | | | | | | | | | | | |
> | **PubMed** | UGBA | | 0.887 | 0.966 | 0.925 | 2.14 | 22.57 | | 0.812 | 0.957 | 0.879 | 0.67 | **11.11** |
> | | GTA | | 0.820 | 0.905 | 0.860 | 3.50 | 55.13 | | 1.000 | 0.976 | 0.988 | 0.80 | **1.90** |
> | | DPGBA | | 0.850 | 0.910 | 0.879 | 3.39 | 8.33 | | 0.785 | 0.850 | 0.816 | 8.78 | **7.65** |
> | | | | | | | | | | | | | | |
>
>
> ## Q1: Experiments on large-scale dataset
> We sincerely apologize for omitting the experimental results on the large-scale dataset OGB-arxiv in the appendix. The relevant results are shown in the table below. We will also add these results to Table 1 in the revised paper. The results demonstrate that on large datasets like OGB-arxiv, our proposed LFD achieves a low attack success rate while maintaining clean accuracy. Furthermore, RA-ASR outperforms other defense methods even after a redirecting attack, indicating that LFD plays a significant role in addressing the surface security problem of defense models.
>
> | Defense | Dataset | | UGBA |  | | | GTA |  | | | DPGBA |  |
> | :--- | :---: | :---: | :---: | :---: | :---: | :---: | :---: | :---: | :---: | :---: | :---: | :---: |
> | | | CA | ASR | RA-ASR | | CA | ASR | RA-ASR | | CA | ASR | RA-ASR |
> | **Clean Model** | **OGB-arxiv** | 60.41 | 11.42 | 20.69 | | 60.38 | 0.84 | 13.58 | | 60.36 | 8.72 | 13.80 |
> | **No Defense** | | 59.32 | 99.79 | 79.55 | | 60.07 | 94.85 | 85.44 | | 59.77 | 93.60 | 84.53 |
> | **OP Model** | | 52.88 | 0.97 | 1.02 | | 58.44 | 0.00 | 2.43 | | 58.60 | 1.07 | 5.48 |
> | **RIGBD** | | 60.43 | 1.02 | 27.28 | | 60.66 | 7.23 | 17.19 | | 61.04 | 5.10 | 20.34 |
> | **DShield** | | 58.95 | 3.09 | 11.18 | | 57.71 | 6.58 | 24.04 | | 58.81 | 2.42 | 16.89 |
> | **LFD(Ours)** | | 59.94 | 0.40 | **4.80** | | 58.89 | 6.29 | **12.87** | | 59.22 | 3.61 | **9.86** |
> | | | | | | | | | | | | | |
>
>
> ## Q2: Further analysis of Table 2
> We are truly impressed by your keen observation regarding Table 2. We carefully analyzed this phenomenon and ultimately concluded that the presence of the llfc_loss term explicitly forces the model to move away from the backdoor space in the feature space, in order to learn more robust benign features. In contrast, when training with only clean_loss, the model is initialized with backdoor weights, so the optimization process may tend to remain in a local optimum, thus limiting the model's generalization ability on clean test data.

---

> ### Author Response · Authors · 2025-11-23
>
> ## Q5: About clean-label setting
> Regarding the limitations mentioned by the reviewer under the clean-label setting, we further explain here. Firstly, our proposed poisoned node detection algorithm based on feature perturbation is currently only applicable to dirty-label. This is because its principle is to use the variance of the prediction results brought about by feature perturbation to distinguish between poisoned and clean nodes. However, under the clean-label setting, poisoned nodes and the target class share the same label, thus limiting the algorithm under this setting. But our method can be **generalized to the clean-label setting using the second-stage feature divergence training based on LLFC**. Specifically, the purpose of the first stage is to identify possible poisoned and clean nodes from the given data. As long as the corresponding sets of nodes of different categories are obtained, a more robust defense model can be trained through the divergence principle of the second stage. Therefore, the poisoned node detection algorithm of the first stage is actually replaceable. To verify the effectiveness of our statement, we selected the poisoned node sets identified by **RIGBD** and **DShield** and trained the defense on the LFD method. The experimental results are shown in the table below. It can be seen that the defense after using LFD is more robust than the defense of these methods themselves.
>
> | Method | Defense | Dataset | | | UGBA |  | | | GTA |  |
> | :--- | :--- | :---: | :---: | :---: | :---: | :---: | :---: | :---: | :---: | :---: |
> | | | | | CA | ASR | RA-ASR | | CA | ASR | RA-ASR |
> | **GCN** | RIGBD | **Cora** | | 81.85 | 0.40 | 40.24 | | 80.37 | 1.07 | 22.75 |
> | | RIGBD-LLFC (Dirty) | | | 83.81 | 0.30 | **31.08** | | 81.11 | 1.82 | **17.15** |
> | | Dshield | | | 82.22 | 0.40 | 86.45 | | 80.37 | 3.32 | 43.68 |
> | | Dshield-LLFC (Dirty) | | | 83.12 | 0.00 | **44.62** | | 80.37 | 3.11 | **33.07** |
> | | | | | | | | | | | |
> | | RIGBD | **PubMed** | | 83.51 | 2.14 | 22.57 | | 84.32 | 3.50 | 55.13 |
> | | RIGBD-LLFC (Dirty) | | | 83.74 | 3.27 | **10.21** | | 83.29 | 3.66 | **49.10** |
> | | Dshield | | | 83.87 | 2.97 | 86.41 | | 83.61 | 0.80 | **1.83** |
> | | Dshield-LLFC (Dirty) | | | 83.54 | 2.56 | **28.14** | | 83.54 | 0.80 | 1.84 |
> | | | | | | | | | | | |
>
> Furthermore, the **DShield** method can also be implemented with a clean-label setting. Therefore, we also conducted related experiments using the GCBA attack. As can be seen from the results in the table below, for clean-label backdoor attacks, LFD training also outperforms its original performance. In conclusion, although our method cannot be directly used with a clean-label setting, it can serve as a **general training paradigm**, directly applied after various poisoned node identification algorithms to further improve the robustness of the defense model.
>
> | Defense | Dataset | | | GCBA |  |
> | :--- | :---: | :---: | :---: | :---: | :---: |
> | | | | CA | ASR | RA-ASR |
> | Dshield | **Cora** | | 82.59 | 2.76 | 52.11 |
> | Dshield-LLFC (Clean) | | | 81.89 | 2.81 | **15.54** |
> | | | | | | |
> | Dshield | **PubMed** | | 84.23 | 3.07 | 37.66 |
> | Dshield-LLFC (Clean) | | | 83.96 | 3.04 | **19.63** |
> | | | | | | | | | | |
>
>
> ## Conclusion
> We hope our response above has resolved your issue. If you have any further questions or details about the paper, we look forward to your suggestions. Some visualization results have been directly updated in Appendix of the revised paper; please feel free to view them if needed. We are currently working diligently to revise and improve the overall content and layout of the paper and will upload the final version as soon as possible.
>
> Best,
>
> Authors

---

> ### Author Response · Authors · 2025-11-27
> **Sincerely looking forward to your reply!**
>
> Dear Reviewer **LrTi**,
>
> We hope this message finds you well.
>
> We have carefully addressed your previous comments and made substantial revisions to our manuscript, as detailed in the responses and the revised PDF file (Text marked in blue).
>
> As the discussion period is approaching its deadline, we look forward to your feedback on the revised submission at your earliest convenience. Your insights and opinions are crucial for us to further improve the quality of the manuscript, and we greatly value the opportunity to continue communicating with you.
>
> Thanks again for your time and effort!
>
> Best,
>
> Authors

---

### Official Review · Reviewer_6y4r · 2025-11-01

**Soundness:** 2
**Presentation:** 2
**Contribution:** 2
**Rating:** 4
**Confidence:** 3

**Summary:**

This paper presents a systematic robustness analysis of existing graph backdoor defense methods. The authors identify a critical weakness that many defenses that appear to be effective only achieve superficial security. This security is easily broken by a simple retuning attack, where the defended model is fine-tuned for just a few epochs on a small amount of poisoned data, causing the ASR to rebound sharply. To explain this fragility, the authors leverage Layer-wise Linear Feature Connectivity  to show that "purified" models remain highly connected to the original backdoored model in the feature space. Building on this insight, they propose a novel defense strategy, Layer-wise Feature Divergence , which explicitly maximizes the feature-space divergence between the defense model and the backdoored model during retraining. Extensive experiments across multiple datasets and attacks demonstrate that LFD achieves significantly more robust defense compared to existing state-of-the-art methods.

**Strengths:**

1. This is the first work to systematically reveal and analyze the superficial security problem in graph backdoor defenses. The finding that defenses can be easily broken is significant for the community and highlights a critical blind spot in current research.

2. The use of LLFC as an analytical tool to explain the root cause of the fragility is innovative and well-motivated. It moves beyond simple parameter-space analysis and provides a convincing, feature-space perspective on why backdoor knowledge persists.

3.  The proposed LFD defense is a direct and elegant solution to the identified problem. The comprehensive experiments across multiple datasets and attacks, along with thorough ablation studies, robustly validate its effectiveness and the contribution of each component.

**Weaknesses:**

1. As noted in the limitations, the work currently focuses only on dirty-label attacks. Furthermore, the practical threat model of the Retuning Attack requires clarification. The paper's background positions RA as an attacker's action, yet the scenario often implies a defender using it for detection. This ambiguity weakens the motivation. Moreover, if an attacker has the capability to perform RA, it is unclear why they wouldn't simply mount a new backdoor attack from scratch to achieve similar or better effects, which questions the practical necessity of RA.

2. The core mechanism of simulating backdoor activation without knowing the true trigger is not sufficiently justified. The paper claims that adding random noise to node features can help distinguish poisoned nodes, but the theoretical or empirical basis for why this perturbation reliably mimics the effect of the true, structured trigger is lacking. Experimental validation specifically supporting this design choice is needed to confirm its validity over simple random sampling.

3. The description of the poisoned node detection algorithm and the captions for key figures lack clarity, hindering reproducibility. Additionally, the computational overhead of LFD, which requires forward passes through three models, is not discussed, which is a practical concern for adoption.

**Questions:**

1. The paper's core vulnerability analysis relies on the Retuning Attack. However, the threat model seems ambiguous. Is RA an action performed by an attacker who has obtained a purified model, or is it a diagnostic tool for the defender? If it's an attacker's action, what is the practical advantage of RA over simply training a new backdoored model, given that both require poisoning the training data?

2. In your poisoned node detection method, you use random feature perturbation on all nodes to simulate potential trigger effects. What is the theoretical or empirical evidence that this general perturbation effectively activates the specific, often structure-based, backdoor features implanted by attacks like GTA or UGBA? Are there ablation studies or comparisons that demonstrate the superiority of this specific perturbation strategy?

---

> ### Author Response · Authors · 2025-11-23
>
> Dear Reviewer **6y4r**,
>
> We sincerely appreciate your valuable and constructive feedback, which have helped us further improve the quality and clarity of the manuscript. We have carefully considered every comment, and below is our point-by-point response:
>
> ## W1 & Q1: About retuning attack (RA) and clean-label setting
> The reviewer's questions regarding the definition of RA are very precise, and we apologize for any confusion caused by the unclear explanation in the motivation section. RA is a diagnostic tool used by defenders to evaluate the robustness of defense models. Specifically, when a defender proposes a defense method, to verify whether the method truly eliminates the backdoor, the defender simulates a **"secondary attacker"** with a small amount of poisoned data to see if they can reactivate the backdoor. RA analyzes the backdoor's residual issues by comparing the difference in ASR between subsequent attacks. Of course, your point that training a backdoor model from scratch is more effective for attackers is entirely correct, but this is not the purpose or setting of RA. We use RA to reveal significant security vulnerabilities in existing defense methods from the defender's perspective.
>
> Regarding the limitations mentioned by the reviewer under the clean-label setting, we further explain here. Firstly, our proposed poisoned node detection algorithm based on feature perturbation is currently only applicable to dirty-label. This is because its principle is to use the variance of the prediction results brought about by feature perturbation to distinguish between poisoned and clean nodes. However, under the clean-label setting, poisoned nodes and the target class share the same label, thus limiting the algorithm under this setting. But our method can be **generalized to the clean-label setting using the second-stage feature divergence training based on LLFC**. Specifically, the purpose of the first stage is to identify possible poisoned and clean nodes from the given data. As long as the corresponding sets of nodes of different categories are obtained, a more robust defense model can be trained through the divergence principle of the second stage. Therefore, the poisoned node detection algorithm of the first stage is actually replaceable. To verify the effectiveness of our statement, we selected the poisoned node sets identified by **RIGBD** and **DShield** and trained the defense on the LFD method. The experimental results are shown in the table below. It can be seen that the defense after using LFD is more robust than the defense of these methods themselves.
>
> | Method | Defense | Dataset | | | UGBA | | | | GTA|  |
> | :--- | :--- | :---: | :---: | :---: | :---: | :---: | :---: | :---: | :---: | :---: |
> | | | | | CA | ASR | RA-ASR | | CA | ASR | RA-ASR |
> | **GCN** | RIGBD | **Cora** | | 81.85 | 0.40 | 40.24 | | 80.37 | 1.07 | 22.75 |
> | | RIGBD-LLFC (Dirty) | | | 83.81 | 0.30 | **31.08** | | 81.11 | 1.82 | **17.15** |
> | | Dshield | | | 82.22 | 0.40 | 86.45 | | 80.37 | 3.32 | 43.68 |
> | | Dshield-LLFC (Dirty) | | | 83.12 | 0.00 | **44.62** | | 80.37 | 3.11 | **33.07** |
> | | | | | | | | | | | |
> | | RIGBD | **PubMed** | | 83.51 | 2.14 | 22.57 | | 84.32 | 3.50 | 55.13 |
> | | RIGBD-LLFC (Dirty) | | | 83.74 | 3.27 | **10.21** | | 83.29 | 3.66 | **49.10** |
> | | Dshield | | | 83.87 | 2.97 | 86.41 | | 83.61 | 0.80 | **1.83** |
> | | Dshield-LLFC (Dirty) | | | 83.54 | 2.56 | **28.14** | | 83.54 | 0.80 | 1.84 |
> | | | | | | | | | | | |
>
> Furthermore, the **DShield** method can also be implemented with a clean-label setting. Therefore, we also conducted related experiments using the GCBA attack. As can be seen from the results in the table below, for clean-label backdoor attacks, LFD training also outperforms its original performance. In conclusion, although our method cannot be directly used with a clean-label setting, it can serve as a **general training paradigm**, directly applied after various poisoned node identification algorithms to further improve the robustness of the defense model.
>
> | Method | Defense | Dataset | | | GCBA |  |
> | :--- | :--- | :---: | :---: | :---: | :---: | :---: |
> | | | | | CA | ASR | RA-ASR |
> | **GCN** | Dshield | **Cora** | | 82.59 | 2.76 | 52.11 |
> | | Dshield-LLFC (Clean) | | | 81.89 | 2.81 | **15.54** |
> | | | | | | | |
> | | Dshield | **PubMed** | | 84.23 | 3.07 | 37.66 |
> | | Dshield-LLFC (Clean) | | | 83.96 | 3.04 | **19.63** |
> | | | | | | | | | | | |

---

> ### Author Response · Authors · 2025-11-23
>
> ## W2 & Q2: About poisoned node detection algorithm
> Regarding the reviewer‘s question about the poisoned node detection algorithm, we provide further explanation and supplementation here. Our algorithm primarily utilizes the difference in the impact of random perturbations of neighbor features on the prediction results to distinguish between poisoned and clean nodes. For clean nodes, the prediction result is highly dependent on the feature aggregation of benign neighbors; therefore, changes in the features of surrounding neighbors will significantly affect the prediction result. For poisoned nodes, the prediction result is highly dependent on triggers, and the mapping from triggers to target labels is generally overfitted. Therefore, even if a certain perturbation is added to the neighbor triggers, the change in the prediction result is not significant. We emphasize that the purpose of using feature perturbation is not to simulate triggers for different attack methods (such as structural perturbation), but to identify nodes that are not sensitive to neighbor feature perturbation. To intuitively illustrate the effectiveness of the poisoned node detection algorithm, we supplemented the following experiments. The overall defense mode of the graph backdoor defense method **RIGBD** is the same as our **LFD**, mainly identifying poisoned nodes in the first stage. Therefore, we selected three metrics **recall**, **precision**, and **F1-score** to compare the recognition performance of the two methods. As can be seen from the results in the table below, although our detection algorithm is slightly weaker than RIGBD, in the second stage of defense training, LFD's defense robustness is far superior to RIGBD. This further demonstrates the key role of feature divergence optimization training.
>
> | Datasets | Attacks | | | | RIGBD | |  | | | | LFD | |  |
> | :--- | :--- | :---: | :---: | :---: | :---: | :---: | :---: | :---: | :---: | :---: | :---: | :---: | :---: |
> | | | | Recall | Precision | F1-score | ASR | RA-ASR | | Recall | Precision | F1-score | ASR | RA-ASR |
> | **Cora** | UGBA | | 0.968 | 1.000 | 0.984 | 0.40 | 40.24 | | 0.900 | 1.000 | 0.947 | 0.00 | **15.54** |
> | | GTA | | 0.912 | 0.940 | 0.926 | 1.07 | **22.75** | | 0.875 | 1.000 | 0.933 | 6.10 | 31.07 |
> | | DPGBA | | 1.000 | 1.000 | 1.000 | 6.64 | 23.94 | | 0.869 | 0.914 | 0.891 | 9.16 | **21.91** |
> | | | | | | | | | | | | | | |
> | **OGB-arxiv**| UGBA | | 0.956 | 0.963 | 0.959 | 1.02 | 27.28 | | 0.852 | 0.910 | 0.880 | 0.40 | **4.80** |
> | | GTA | | 0.849 | 0.903 | 0.875 | 7.23 | 17.19 | | 0.811 | 0.858 | 0.834 | 6.29 | **12.87** |
> | | DPGBA | | 0.905 | 0.939 | 0.922 | 5.10 | 20.34 | | 0.862 | 0.851 | 0.856 | 3.61 | **9.86** |
> | | | | | | | | | | | | | | |
> | **PubMed** | UGBA | | 0.887 | 0.966 | 0.925 | 2.14 | 22.57 | | 0.812 | 0.957 | 0.879 | 0.67 | **11.11** |
> | | GTA | | 0.820 | 0.905 | 0.860 | 3.50 | 55.13 | | 1.000 | 0.976 | 0.988 | 0.80 | **1.90** |
> | | DPGBA | | 0.850 | 0.910 | 0.879 | 3.39 | 8.33 | | 0.785 | 0.850 | 0.816 | 8.78 | **7.65** |
> | | | | | | | | | | | | | | |
>
> Furthermore, the reviewer mentioned a comparison with "simple random sampling." We added some experiments to demonstrate the effectiveness of the proposed heuristic detection algorithm. Specifically, **LFD-random** represents randomly selecting nodes with the same number of identified poisoned nodes as poisoned nodes, while **LFD-none** represents normal fine-tuning training without distinguishing between poisoned and clean nodes. The table below shows the CA, ASR, and RA-ASR for the three cases. The results show that the standard LFD performs significantly better than both variants across all three metrics.
>
> | Datasets | Attacks | | | LFD |  | | | LFD-random |  | | | LFD-none |  |
> | :--- | :--- | :---: | :---: | :---: | :---: | :---: | :---: | :---: | :---: | :---: | :---: | :---: | :---: |
> | | | | CA | ASR | RA-ASR | | CA | ASR | RA-ASR | | CA | ASR | RA-ASR |
> | **Cora** | UGBA | | 83.70 | 0.00 | **15.54** | | 82.59 | 40.95 | 90.84 | | 84.81 | 54.58 | 84.10 |
> | | GTA | | 83.70 | 6.10 | **31.07** | | 80.00 | 20.56 | 56.97 | | 83.33 | 41.83 | 33.55 |
> | | DPGBA | | 83.33 | 9.16 | **21.91** | | 84.44 | 19.12 | 32.97 | | 83.32 | 29.32 | 31.66 |
> | | | | | | | | | | | | | | |
> | **OGB-arxiv**| UGBA | | 59.94 | 0.40 | **4.80** | | 61.62 | 14.60 | 32.41 | | 61.40 | 50.27 | 47.22 |
> | | GTA | | 58.89 | 6.29 | **12.87** | | 61.01 | 16.44 | 27.13 | | 61.16 | 19.71 | 29.22 |
> | | DPGBA | | 59.22 | 3.61 | **9.86** | | 60.83 | 10.87 | 26.99 | | 61.79 | 11.55 | 27.27 |
> | | | | | | | | | | | | | | |

---

> ### Author Response · Authors · 2025-11-23
>
> ## W3: The computational overhead of LFD
> The reviewer considered the computational cost analysis essential, and we fully agree with the suggestion. We have added experiments, and the relevant results are shown in the table below. The table primarily displays the training and testing times for different defense methods, as well as the maximum memory usage during training. Although our LFD method outperforms the standard model and RIGBD in terms of training time and memory usage, these costs can be considered one-time expenses in the defense process. Once LFD is trained, its computational cost during the inference phase is close to that of the standard GCN model and lower than other defense methods.
>
> | Defense | Dataset | UGBA | | | GTA | | | DPGBA | |
> | :--- | :---: | :---: | :---: | :---: | :---: | :---: | :---: | :---: | :---: |
> | | | Train/Test_time | Memory | | Train/Test_time | Memory | | Train/Test_time | Memory |
> | **Clean Model** | **Cora** | 0.97s/0.65s | 0.0378GB | | 1.00s/0.64s | 0.0452GB | | 1.03s/0.65s | 0.0377GB |
> | **RIGBD** | | 4.00s/1.86s | 0.3295GB | | 5.96s/1.88s | 0.3427GB | | 3.53s/1.04s | 0.3581GB |
> | **DShield** | | 108.00s/2.80s | 1.3794GB | | 115.51s/3.81s | 1.6733GB | | 67.65s/2.76s | 1.5937GB |
> | **LFD(Ours)** | | 9.8s/0.79s | 0.3454GB | | 7.14s/0.62s | 0.3825GB | | 5.69s/0.79s | 0.4671GB |
> | | | | | | | | | | |
> | **Clean Model** | **PubMed** | 3.04s/1.86s | 0.1061GB | | 3.13s/1.84s | 0.1154GB | | 3.27s/1.80s | 0.1306GB |
> | **RIGBD** | | 14.47s/2.55s | 0.4616GB | | 16.35s/3.66s | 0.4822GB | | 17.46s/3.14s | 0.5186GB |
> | **DShield** | | 126.84s/5.04s | 1.5477GB | | 126.48s/5.75s | 1.6943GB | | 116.45s/5.50s | 1.6854GB |
> | **LFD(Ours)** | | 19.56s/2.06s | 0.4983GB | | 19.52s/2.97s | 0.5559GB | | 19.30s/2.22s | 0.5560GB |
> | | | | | | | | | | |
> | **Clean Model** | **OGB-arxiv**| 15.13s/51.46s | 1.0222GB | | 15.12s/64.36s | 1.0247GB | | 15.12s/61.43s | 1.0231GB |
> | **RIGBD** | | 21.13s/74.40s | 2.8764GB | | 21.37s/71.48s | 2.7842GB | | 20.23s/73.76s | 2.7628GB |
> | **DShield** | | 920.39s/85.19s | 4.9430GB | | 884.26s/96.16s | 4.9484GB | | 953.05s/117.26s | 4.9362GB |
> | **LFD(Ours)** | | 25.76s/61.62s | 2.9437GB | | 32.74s/67.22s | 2.9792GB | | 24.17s/63.74s | 2.8241GB |
> | | | | | | | | | | |
>
>
> ## Conclusion
> We hope our response above has resolved your issue. If you have any further questions or details about the paper, we look forward to your suggestions. Some visualization results have been directly updated in Appendix of the revised paper; please feel free to view them if needed. We are currently working diligently to revise and improve the overall content and layout of the paper and will upload the final version as soon as possible.
>
> Best,
>
> Authors

---

> ### Author Response · Authors · 2025-11-27
> **Sincerely looking forward to your reply!**
>
> Dear Reviewer **6y4r**,
>
> We hope this message finds you well.
>
> We have carefully addressed your previous comments and made substantial revisions to our manuscript, as detailed in the responses and the revised PDF file (Text marked in blue).
>
> As the discussion period is approaching its deadline, we look forward to your feedback on the revised submission at your earliest convenience. Your insights and opinions are crucial for us to further improve the quality of the manuscript, and we greatly value the opportunity to continue communicating with you.
>
> Thanks again for your time and effort!
>
> Best,
>
> Authors

---

### Author Response · Authors · 2025-12-01
**Overall response to Area Chair**

Dear **Area Chair**,

We hope this message finds you well.

By the time the leak occurred on a large scale, we had not received any further discussion from the reviewers. In our response, we have fully addressed all the reviewers' questions, provided additional experimental results and supplementary explanations. Therefore, we sincerely hope that the AC will carefully review our detailed responses to the initial review comments from the four reviewers.

For your convenience, we have summarized all the responses. Specifically, regarding the issues raised by multiple reviewers concerning the poisoned node detection algorithm, performance on large-scale dataset, the computational overhead and further theoretical analysis of the method LFD, we have supplemented our work with corresponding experimental results and more convincing arguments. For individual reviewer's issues regarding the ablation analysis of hyperparameters, the impact of feature divergence at different layers, the quantitative relationship between LLFC and RA-ASR, the visualization analysis, confusions about the retuning attack and the backdoor model, we have also provided experimental analysis and detailed explanations for each. Furthermore, regarding the limitations of the clean-label setting mentioned in our Limitation section, we have provided all reviewers with a more detailed analysis and an achievable solution.

Based on the reviewers' suggestions, we have also made corresponding revisions to the revised PDF, which are marked in blue text. Specifically, we added OGB-arxiv experimental results to Section 6.2 (Table 1). In Section 6.3, we added ablation experiments on feature divergence at different layers (Table 2), and a visualization result of the node feature distribution of defense methods (Figures 3 and 4). In Section 6.4, we added hyperparameter analysis of the optimization terms (Figure 5). In the Appendix, we added statistics on the datasets (Table 4), a quantitative relationship analysis between LLFC and RA-ASR (Appendix C), supplementary results of the hyperparameter analysis (Appendix D), comparison results of poisoned node detection algorithms (Appendix E), computational cost of LFD (Appendix F), and supplementary analysis under the clean-label setting (Appendix G).

We hope our overall responses could resolve your issues, and thanks again for your time and effort during the review process.

Sincerely,
**Authors**

---

### Meta-Review · Area_Chair_R7o7 · 2026-01-07

**Summary:**

In the rebuttal, the authors provided extensive new experiments and clarifications, including the problem setup, motivation, scalability to clean-label and black-box attacks, and computational cost, which address several key concerns. However, some core limitations regarding the work's limited novelty, heuristic designing and lack of theoretical analysis, and its modest technical contribution are not fully addressed. Taking all reviewer comments into account, while the work has clear merit, it remains insufficient for acceptance in its current form.

**Reviewer Concerns:**

Some core limitations regarding the work's limited novelty, heuristic designing and lack of theoretical analysis, and its modest technical contribution are not fully addressed.

**Reviewer Scores:**

Reviewers may not change their scores.

---

### Decision · Program_Chairs · 2026-01-26

Reject